# `Flex-Act`: Why Learn when you can Pick?

**Ramnath Kumar**                                                    *ramnathkumar181@gmail.com*
*University of California Los Angeles.*

**Kyle Ritscher**                                                      *kritscher@g.ucla.edu*
*University of California Los Angeles.*

**Junmin Zhu**                                                         *judyz@g.ucla.edu*
*University of California Los Angeles.*

**Lawrence Liu**                                                       *lawrencerliu@g.ucla.edu*
*University of California Los Angeles.*

**Cho-Jui Hsieh**                                                      *chohsieh@cs.ucla.edu*
*University of California Los Angeles, Arena.*

**Reviewed on OpenReview:** *https://openreview.net/forum?id=HQGis83pM2*

## Abstract

Learning activation functions has emerged as a promising direction in deep learning, allowing networks to adapt activation mechanisms to task-specific demands. In this work, we introduce a novel framework that employs the Gumbel-Softmax trick to enable discrete yet differentiable selection among a predefined set of activation functions during training. Our method dynamically learns the optimal activation function independently of the input, thereby enhancing both predictive accuracy and architectural flexibility. Experiments on synthetic datasets show that our model consistently selects the most suitable activation function, underscoring its effectiveness. These results connect theoretical advances with practical utility, paving the way for more adaptive and modular neural architectures in complex learning scenarios.

## 1 Introduction

While modern deep learning architectures have revolutionized domains spanning vision, language, and control, a surprisingly under-explored axis in this progression lies in the choice and design of activation functions. These pointwise non-linearities not only endow neural networks with the capacity to model complex functions, but also profoundly influence trainability, signal propagation, and generalization behavior (Nair & Hinton, 2010; Nwankpa et al., 2018; Pedamonti, 2018; Subramanian et al., 2024). As we scale models to unprecedented depths and widths, activation functions become increasingly critical mediators of inductive bias, information flow, and learning dynamics. Yet, in most empirical pipelines, the activation function is treated as a fixed hyperparameter-often defaulting to ReLU, GELU, or their minor variants. Recent work on scaling laws (Kaplan et al., 2020) has identified predictable trends in model performance with respect to compute, data, and parameter count. These observations have shaped the design of large-scale transformers and vision architectures, spurring innovations in initialization (Saxe et al., 2014; He et al., 2015), normalization (Ba et al., 2016; Zhang et al., 2019), and architecture (Vaswani et al., 2017; Dosovitskiy et al., 2021). However, activation functions remain a surprisingly static component within this dynamic progress of neural networks, and their ongoing research. When scaled to hundreds of layers, signal fidelity and gradient flow hinges on the choice of non-linearity: whether activations preserve variance across layers (Glorot & Bengio, 2010), whether they induce saturation or sparsity (Maas et al., 2013), and whether their derivatives yield favorable Hessian spectra (Pennington et al., 2017) - all of these directions emerge as pivotal factors in the success of large

models. The spectral properties of activations also interact with the neural tangent kernel (Murray et al., 2023), further motivating careful activation design beyond simple empirical heuristics.

From the perspective of representational power, activation functions define the functional class accessible to the network. A poor choice can lead to shattered gradients, and unstable dynamics (Balduzzi et al., 2017). For instance, the introduction of Swish and GELU brought improved performance through smoother transitions and better gradient characteristics (Ramachandran et al., 2018; Hendrycks & Gimpel, 2016), yet these improvements were often empirical and lacked a deep mechanistic understanding in high-dimensional, deep neural networks. Crucially, in the context of pretraining and finetuning large models, activation functions also affect the scaling of logits and their calibration, with downstream effects on optimization dynamics, learning rate schedules, and generalization (Zhang et al., 2017).

In this work, we argue for a re-examination of activation function design in the context of deep scaling. Conventional activations such as ReLU (Nair & Hinton, 2010), GELU (Hendrycks & Gimpel, 2016), and Swish (Ramachandran et al., 2018) are statically defined, applied identically across layers, tasks, and data regimes. However, this static design paradigm limits adaptability and ignores the growing evidence that no single non-linearity is universally optimal across depth or domain. This mismatch becomes especially pronounced in deep networks where representational demands evolve layer-wise.

**Motivation.** Standard neural networks commit to a fixed activation function before training begins. While some functions generalize well across tasks, no single activation performs optimally across the spectrum of functional regimes encountered in practice. This inflexibility becomes particularly limiting when one wishes to deploy the *same architecture* across multiple datasets or target mappings, where the most effective non-linearity may differ. For example, a task involving bounded, saturating behavior may favor a Sigmoid or Tanh, while another may benefit from the sparsity and unboundedness of ReLU. Recent works have proposed parameterized or learnable activations that morph during training (He et al., 2015; Tavakoli et al., 2021), or even architectures that learn the entire activation function as a neural approximator (Liu et al., 2025). While expressive, these methods often come with higher optimization complexity, reduced interpretability, and the need to commit to a specific parameterization class. Instead, we explore a simpler but powerful alternative: allow each layer to *choose* from a fixed basis of standard activation functions. The selection is learned dynamically during training, and is jointly optimized with model weights. This enables a single network-without any manual hyperparameter tuning or function sweeping-to recover the appropriate non-linearity for the task at hand. We demonstrate this in synthetic regression tasks where the ground-truth activation is known but hidden: our model discovers and adopts the correct function without needing to be retrained across tasks.

**Overview and Evaluation.** We introduce `Flex-Act`, a framework for discrete activation selection based on Gumbel-Softmax routing. Each layer selects its activation from a small set of candidate functions, with selection probabilities learned through backpropagation. To mitigate biases toward unbounded activations, we introduce a novel gradient-norm-based regularizer that aligns routing choices with functional suitability. `Flex-Act` is evaluated in settings where the true activation function governing the target is known, and we compare its performance against fixed activation baselines and parameterized methods. Our results show that `Flex-Act` consistently adapts to the correct activation function without requiring multiple models, activation sweeps, or architectural tuning enabling plug-and-play generalization across diverse function classes. This capability makes it a promising tool for robust, general-purpose deep learning pipelines.

**Contributions.** This paper introduces a new paradigm for non-linearity design in deep networks through *discrete activation selection*, where each layer dynamically routes its input through one of several candidate activation functions. Our core contributions are as follows:

- **Discrete Activation Routing via Gumbel-Softmax.** We present a novel mechanism for layer-wise selection of activation functions using the Gumbel-Softmax reparameterization trick, allowing discrete function choices to be optimized end-to-end via gradient descent. (see Section 3.2)
- **Gradient Derivations for Discrete Non-Linearities.** We analytically derive the backpropagation equations for networks employing discrete activation routing, overcoming challenges posed by non-differentiable selection. (see Section 3.2)

- **Empirical Gains in Deep Settings.** Our approach consistently improves accuracy and training stability over fixed or parameterized activation baselines, particularly in very deep architectures where functional diversity across layers is critical. `Flex-Act` with gradient-norm regularisation achieves more reliable activation selection than unregularised discrete routing. (see Section 4)
- **Interpretable and Modular Non-Linearity Design.** By exposing activation selection as a discrete, learnable variable, our framework enables new avenues for analyzing functional usage across depth, offering both interpretability and architectural flexibility. (see Section 5)

## 2 Related Work

Activation functions are fundamental components of neural networks, providing non-linear transformations that are necessary for approximating complex functions. Their design and choice significantly influence the performance and trainability of deep learning models. This section reviews the evolution of activation function research, from traditional functions to parameterized approaches. We also note there is significant overlap with the literature in online learning and bandits. Throughout this section, we highlight how our work differs by focusing on discrete activation selection tailored to layer-specific requirements. The study of activation functions in neural networks spans from classical hand-crafted forms to recent efforts in adaptive and learnable designs. In this section, we organize prior work into: (i) Traditional Fixed Activations, (ii) Parameterized Activation Functions, and (iii) Discrete or Learnable Function Routing.

**Traditional Fixed Activation Functions.** Early neural networks relied on activation functions such as Sigmoid and Tanh, which introduced the essential non-linearity required to model complex relationships. These functions were instrumental in enabling neural networks to approximate arbitrary functions. However, as networks grew deeper, their limitations became evident. Both Sigmoid and Tanh functions suffered from the vanishing gradient problem, where gradients become increasingly small as they propagate through layers. This hindered the ability of earlier layers to update effectively during training as discussed in Nair & Hinton (2010). ReLU enabled deeper networks by improving gradient flow and computational efficiency (Nair & Hinton, 2010). However, ReLU itself introduced issues such as the "dying ReLU" problem, where neurons become inactive and cease to contribute to learning if the inputs become negative. Variants such as Leaky ReLU (Maas et al., 2013) and ELU (Exponential Linear Unit) (Clevert et al., 2016) were proposed to mitigate these issues, allowing small, non-zero gradients for negative inputs and ensuring smoother transitions. Another significant advancement came with GELU (Gaussian Error Linear Unit) (Hendrycks & Gimpel, 2016), a function that combines the strengths of ReLU and Sigmoid. GELU applies a smooth, differentiable approximation of a gating mechanism, blending linear and non-linear behaviors. It has shown particular effectiveness in natural language processing models like BERT (Devlin et al., 2019) and transformer architectures (Vaswani et al., 2017). GELU ensures smoother gradients and better convergence properties, making it a popular choice in many state-of-the-art architectures. Despite these innovations, traditional activation functions remain fixed across layers and tasks, limiting their adaptability to diverse requirements. Our work builds on the recognition that a single, fixed activation function may not be optimal for all layers in a deep network. Instead of relying on fixed functions or parameterizing a single family, we propose a framework where layers dynamically select the most appropriate activation function from a predefined set, addressing layer-specific (primarily penultimate layer) needs more effectively. This approach is akin to hypermater tuning, but instead of training multiple models, we let the current model learn an optimal function map.

**Parameterized Activation Functions.** To overcome the limitations of fixed activation functions, researchers have focused on parameterized activation functions. It introduces learnable parameters that can be adapted during training, which enables the network to optimize the activation function for specific tasks. This adaptability allows networks to better capture data characteristics, improving convergence rates, expressiveness, and task-specific performance (Goyal et al., 2020). For example, Parametric ReLU (PReLU) extends the Leaky ReLU's flexibility by making the slope parameter $p$ trainable. It is defined as:

$$\mathrm{PReLU}(x) = \begin{cases} x & \text{if } x > 0, \\ px & \text{if } x \leq 0, \end{cases}$$

in the output range of $(-\infty, \infty)$. Like Leaky ReLU, this approach mitigates the dead neuron problem, allowing the network to adaptively determine the gradient flow for negative inputs and improve convergence (Dubey et al., 2022). In general, by adjusting for parametrization, PReLU reduces sensitivity to initialization and increases performance across tasks. Similarly, SPLASH (Simple Piecewise Linear and Adaptive with Symmetric Hinges) (Tavakoli et al., 2021) uses a piecewise linear activation to enhance flexibility. By employing symmetry, grounding, and fixed hinge points, SPLASH simplifies the learning process by reducing the parameter space from $3S + 2$ to $S + 1$, enabling faster optimization and improved generalization. The activation of a hidden unit $h(x)$ in SPLASH is formulated as $S + 1$ max functions with $S$ symmetric offsets, where $S$ is an odd number, and one of the offsets is zero:

$$h(x) = \sum_{s=1}^{(S+1)/2} a_+^s \max(0, x - b^s) + \sum_{s=1}^{(S+1)/2} a_-^s \max(0, -x - b^s),$$

where the learned parameters $a_+^s, a_-^s$ determine the slope of each line segment, and the hinge locations $b^s, -b^s$ are fixed. Despite having fewer parameters, though, SPLASH remains capable of approximating a wide range of non-linear functions, and has demonstrated effectiveness in tasks requiring robust generalization and resistance to adversarial attacks. For instance, in classification problems, SPLASH has outperformed traditional activations like ReLU and Leaky ReLU in terms of both accuracy and adversarial robustness (Tavakoli et al., 2021). The symmetry and continuity of the activation function result in smoother decision boundaries, enhancing the model's resilience to adversarial perturbations. Finally, SWISH (Ramachandran et al., 2018) blends linear and sigmoid components, providing smoother transitions between activations. The SWISH activation function is defined as:

$$\text{SWISH}(x; \beta) = x \cdot \sigma(\beta x),$$

where $\sigma(x) = \frac{1}{1+e^{-x}}$ is the sigmoid function, and $\beta$ is a constant or trainable parameter that controls the degree of non-linearity. When $\beta \to 0$, SWISH behaves like a linear activation function, and as $\beta \to \infty$, it becomes similar to the ReLU activation (Ramachandran et al., 2018). As such, SWISH can be loosely viewed as a smooth function that nonlinearly interpolates between the linear and the ReLU functions. One of the key advantages that differentiates SWISH from other activation functions is its smoothness and non-monotonicity. The differentiability of SWISH ensures that gradient-based optimization algorithms can update weights effectively without encountering abrupt changes, as often seen in ReLU due to its non-smooth nature at $x = 0$. The non-monotonicity introduced by the sigmoid term also allows SWISH to capture more complex relationships in the data compared to monotonic functions like ReLU (Nair & Hinton, 2010). Finally, the inclusion of the trainable parameter $\beta$ allows the network to dynamically adjust the activation function during training. This flexibility ensures SWISH can adapt to the characteristics of the data and the architecture of the network, potentially resulting in better optimization and generalization. In experiments, SWISH demonstrated superior performance across various deep learning tasks, particularly in ImageNet with architectures such as MobileNet and Inception-ResNet (Szegedy et al., 2017).

**General Parametric Activation Function.** Hu et al. (2022) takes the most general approach to researching parametrized activation functions. They study a model which parametrizes any "traditional" activation function $\sigma(\cdot)$, defining the layer-specific function as:

$$\sigma_i(a_i, b_i, c_i, d_i, z) = b_i \sigma(a_i z + c_i) + d_i,$$

where $z = wx + b$ denotes the weighted sum of inputs, including the bias term. These approaches significantly improve performance by tailoring activation behavior to specific tasks, at a relatively cheap cost of only a few parameters per layer. However, parameterized activations assume that a single functional form, albeit adjustable, can optimally serve all layers in a network. This assumption limits their ability to capture the diverse requirements of different layers. Layers closer to the input may require smoother transformations for feature extraction, while deeper layers may benefit from sharper non-linearities for decision boundaries. Our work differs by addressing these layer-specific needs directly, providing greater flexibility without the complexity of designing or training parameterized families.

**Discrete Function Selection and Activation Routing.** The most similar approach to our work would be that of Manessi & Rozza (2018), who consider linear combinations of multiple activations, enabling smooth

transitions and simplifying optimization. Our approach could actually be seen as simply a discrete restriction of their method. However, while effective at addressing layer-specific needs, their method does not provide the interpretability of ours, as the resulting activation is a weighted blend rather than a distinct choice. Furthermore, the blend was restricted to the functions: Identity, ReLU, and Tanh with the ReLU function often receiving a larger score. We experience this bias ourselves, and discovered it is due to the magnitude differences of the activations, where ReLU's unbounded nature causes it to be preferred in optimization procedures, regardless of the data structure. However, unlike Manessi & Rozza (2018), we were able to discover and address this bias due to the interpretability of our method and the structure of our experiments. After our correction, our model successfully picks Sigmoid or Tanh if they are more suitable representations, while such validation is not available in prior literature.

A significant parallel to our work involves parameterizing activation functions as dynamic systems governed by differential equations. Differential Equation Units (DEUs) enable neurons to learn unique non-linearities by solving second-order linear ordinary differential equations (ODEs), providing an uncountably large functional space that can capture oscillatory or harmonic behaviors beyond the reach of fixed activations (Torkamani et al., 2019). Similarly, Neural Memory ODEs (nmODE) improve upon standard Neural ODEs by utilizing memory neurons to establish non-linear mappings toward global attractors, enhancing robustness and predictive performance (Yi, 2023). While these approaches offer exhaustive expressive power, they introduce high-dimensional parameter spaces that are often difficult to optimize and lack the direct interpretability of canonical functions. In contrast, `Flex-Act` treats activation selection as a discrete routing problem over a curated set of primitives with well-characterized inductive biases (e.g., ReLU, Sigmoid). By employing the Gumbel-Softmax trick, we maintain end-to-end differentiability while ensuring the learned architecture remains grounded in stable, theoretically-grounded functional forms. This allows `Flex-Act` to achieve a balance between architectural flexibility and training stability, offering a modular efficiency that is more computationally tractable and easier to analyze than continuous ODE-based units.

In summary, prior works have explored static, parameterized, or blended activation mechanisms. Our work departs from these by introducing a modular, interpretable, and efficient framework for activation function selection via discrete routing.

## 3 Background

**Neural Network Preliminaries.** We consider feedforward neural networks as compositional mappings $F : \mathbb{R}^{d_x} \to \mathbb{R}^{d_y}$, defined via a sequence of layers:

$$F(x) = g_N \circ g_{N-1} \circ \cdots \circ g_1(x),$$

where each layer $g_j : \mathbb{R}^{d_{j-1}} \to \mathbb{R}^{d_j}$ consists of an affine transformation followed by a component-wise non-linearity:

$$g_j(z) = \sigma_j(W_j z + b_j), \quad W_j \in \mathbb{R}^{d_j \times d_{j-1}}, \quad b_j \in \mathbb{R}^{d_j}.$$

Here, the activation function $\sigma_j : \mathbb{R} \to \mathbb{R}$ acts coordinate-wise on the vector, i.e., $(\sigma_j(w))_k = \sigma_j(w_k)$. While classical networks use a fixed activation $\sigma_j \equiv \sigma$ across all layers, our work focuses on the flexible and learnable selection of $\sigma_j$ from a finite set, discussed further in Section 3.2.

The network is trained on labeled data $\{(x_i, y_i)\}_{i=1}^m$, minimizing a loss function $\ell : \mathbb{R}^{d_y} \times \mathbb{R}^{d_y} \to \mathbb{R}_+$. The empirical risk objective is:

$$\mathcal{L}(\{W_j, b_j\}) = \sum_{i=1}^m \ell(F(x_i), y_i),$$

optimized via gradient-based methods. The dominant computational workhorse here is backpropagation, which leverages the chain rule to compute gradients efficiently (Rumelhart et al., 1986). This backpropagation process depends on the form and differentiability of each $\sigma_j$.

**Activation Function Design.** Activation functions introduce the non-linearity required for deep networks to approximate highly complex mappings as compiled in Dubey et al. (2022). While historically fixed across

all layers, growing model depth and diversity of layer function suggest that more sophisticated, possibly adaptive, designs can offer improved expressiveness and trainability.

Our framework allows each layer to select an activation from a set $\{\sigma^{(1)}, \ldots, \sigma^{(p)}\}$. In this paper, we consider the following activation functions:

- **ReLU**: One of the most popular choices of activation functions is the Rectified Linear Unit (ReLU). It is defined as

$$\text{ReLU}(x) = \max(0, x).$$

  ReLU is computationally efficient and mitigates the vanishing gradient problem for positive inputs. However, the main limitation with ReLU is that all the negative values become zero, and some gradients can be fragile during training, resulting in dead neurons (Agarap, 2019).

- **Sigmoid**: Sigmoid is defined as

$$\text{sigmoid}(x) = \frac{1}{1 + e^{-x}}.$$

  It maps inputs to $[0, 1]$, making it suitable for probabilistic outputs. Despite its interpretability, it faces the "vanishing gradient problem," and the saturating of gradients for large input values poses challenges during training (Boullé et al., 2020).

- **Tanh**: The `Tanh` activation function is defined as

$$\tanh(x) = \frac{e^x - e^{-x}}{e^x + e^{-x}}.$$

  Tanh is a scaled and shifted version of the sigmoid function that maps inputs to $[-1, 1]$, providing zero-centered outputs, which often help with optimization. However, it also suffers from the vanishing gradient problem for large positive or negative inputs, where the function saturates and the derivative approaches zero. It also requires exponentials for computation, which could be slower in training and inference (Dubey et al., 2022).

- **Leaky ReLU**: Leaky ReLU is one of the variations of the ReLU function that addresses the dead neurons problem. It is defined as

$$\text{Leaky ReLU}(x) = \begin{cases} x & \text{if } x > 0, \\ kx & \text{if } x \leq 0, \end{cases}$$

  where $k$ is a small positive constant. Unlike ReLU, Leaky ReLU has a small gradient $k$ for negative inputs, therefore preventing the dead neuron problem. It also mitigates the gradient saturation problem for positive inputs. However, Leaky ReLU requires tuning another hyperparameter, and it does not always outperform the original ReLU activation function (Boullé et al., 2020; Tavakoli et al., 2021).

- **Identity**: The identity function is defined as

$$f(x) = x.$$

  The identity function retains linearity, serving as a baseline for testing the network's behavior with non-transformative activations (Dubey et al., 2022).The Identity function serves as an important sanity check in many of our experiments.

This set provides a varied spectrum of behaviors with differing curvature, boundedness, and gradient characteristics. Our model aims to select the most appropriate function at the penulitmate layer by routing dynamically during training.

### 3.1 The Gumbel-Softmax Trick

Optimization involving discrete choices poses a major challenge to gradient-based learning. The Gumbel-Softmax trick (Jang et al., 2017) provides a continuous relaxation to sampling from categorical distributions, allowing for low-variance gradient estimation via reparameterization.

Given class probabilities $\pi_1, \ldots, \pi_k$, the Gumbel-Max trick samples:

$$z = \text{one\_hot}\left(\arg\max_i[\log \pi_i + g_i]\right), \quad g_i \sim \text{Gumbel}(0, 1),$$

where $g_i = -\log(-\log u_i)$ and $u_i \sim \text{Uniform}(0, 1)$.

Replacing $\arg\max$ with a softmax yields the Gumbel-Softmax distribution:

$$y_i = \frac{\exp((\log \pi_i + g_i)/\tau)}{\sum_j \exp((\log \pi_j + g_j)/\tau)},$$

where $\tau$ controls the "sharpness" of selection. As $\tau \to 0$, this approaches a true categorical distribution; higher $\tau$ yields smoother mixtures. The Gumbel-Softmax enables gradient flow through discrete selections, and has been used in areas ranging from generative models (Kusner & Hernández-Lobato, 2016) to channel pruning (Strypsteen & Bertrand, 2021). We employ it to the appropriate activation function suitable for the model without any overhead on hyperparameter search.

**Gradient Estimation.** The key advantage of the Gumbel-Softmax is that it allows gradients with respect to the logits $\log \pi_i$ to be computed via backpropagation. However, as we show in Section 3.3, naive use leads to activation selection biased toward unbounded functions like ReLU. This motivates the normalization strategy introduced in Section 3.3.

### 3.2 Activation Selection via Soft Routing

We now define the proposed model. At each layer $i$, given input $X_{i-1}$, we first compute the affine transformation $h_i = W_i X_{i-1} + b_i \in \mathbb{R}^{d_{\text{out}}}$. Rather than applying a fixed non-linearity, we compute a convex combination over a set of candidate functions at the penulimate layer:

$$X_i = \sum_{j=1}^{p} p_i^{(j)} \cdot \sigma^{(j)}(h_i),$$

where $\mathbf{p}_i \sim \text{GumbelSoftmax}(\log \boldsymbol{\pi}_i, \tau)$. Each $\sigma^{(j)}$ is a pre-defined activation function, and the logits $\boldsymbol{\pi}_i \in \mathbb{R}^p$ are trainable parameters learned jointly with the network weights. As $\tau$ is annealed, the model transitions from exploring blends to committing to specific activations per layer.

### 3.3 Bias Correction via Gradient Normalization

**Scale-Induced Selection Bias.** In preliminary experiments, we observed that the model consistently favored ReLU and Leaky ReLU where the activations have large unbounded outputs regardless of data-specific structure. This arises because backpropagation scales the gradient by the activation value itself, biasing selection toward functions with large magnitudes rather than functions more appropriate for the data regime. We further elaborate on this behavior and motivate our regularizer in Appendix D.

**Weighting with Gradient Norms.** To address the problems observed with the gradient, we introduce a gradient normalization mechanism to prevent this.

We compute the gradient norms of each activation function with respect to the input:

$$g_i = \|\nabla_x \phi_i(h)\|_2, \tag{1}$$

where $h = \bar{\mathbf{X}}\mathbf{W}^\top + \mathbf{b}$.

We convert the negative average gradient norms into pseudo-probabilities using a softmax function:

$$\tilde{p}_i = \frac{\exp(-\bar{g}_i/\lambda)}{\sum_{j=1}^{K} \exp(-\bar{g}_j/\lambda)}, \tag{2}$$

where $\bar{g}_i$ is the average gradient norm for activation function $\phi_i$ over the batch, and $\lambda$ is a scaling factor to adjust the pulling factor. Functions with larger gradients have smaller pseudo-probabilities.

The total loss is then a combination of the primary task loss (e.g., mean squared error for regression) and the Kullback-Leibler divergence between the model's activation probabilities and the pseudo-label probabilities:

$$\mathcal{L} = \mathcal{L}_{\text{task}} + \alpha \mathcal{L}_{\text{KL}}, \tag{3}$$

where

$$\mathcal{L}_{\text{KL}} = \sum_{i=1}^{N} \sum_{k=1}^{p} \tilde{p}_i^{(k)} \log\left(\frac{\tilde{p}_i^{(k)}}{p_i^{(k)}}\right), \tag{4}$$

and $\alpha$ is a weighting coefficient. By including this regularization term, we were able to overcome the problems with the bias towards suboptimal activation functions in our experiments.

## 4  Experiments

In this section, we present an empirical evaluation of `Flex-Act`, our proposed dynamic activation selection framework. We compare it against standard fixed-activation networks and ablate key components of our model, including the use of regularization. Our experiments are designed to isolate the role of activation selection in shallow regression tasks, where an optimal solution is analytically known. This controlled setting allows us to precisely measure convergence, interpretability, and the effect of discrete selection dynamics.

### 4.1  Experimental Setup

We construct a synthetic regression task where the target label is generated via a non-linear activation applied to a single informative input variable, with additional distractor features. Specifically, we generate input vectors $x \in \mathbb{R}^4$, where one dimension $x_1$ is informative, and the remaining are i.i.d. Gaussian noise. The label is given by:

$$y = a(k \, . \, x_1),$$

where $a(\cdot)$ denotes the ground truth activation function, and $k$ is constant, which we arbitrarily set to 5 for the sake of simplicity throughout our experiments. We test five such cases for activation functions: `ReLU`, `Sigmoid`, `Tanh`, `LeakyReLU`, and `Identity`. Our goal is to evaluate whether models can recover this target transformation without any sweep and computational overhead.

We compare the following model variants:

- **`Flex-Act` (Ours)**: A single-layer network with dynamic, layer-wise activation selection using Gumbel-Softmax sampling from five candidate functions. Details are provided in Section 3.

- **Fixed-Activation Networks**: Five networks, each using a fixed activation from the candidate set. This provides an upper bound when the activation matches $a(\cdot)$, and a lower bound when mismatched.

Each model is trained to minimize Mean Squared Error (MSE), and we report both final error and convergence behavior averaged over 5 different seeds. For `Flex-Act` models, we additionally track the evolution of the activation selection probabilities over time for additional insights.

### 4.1.1 Results and Analysis

**Performance Across Ground Truths.** Table 1 reports the mean and standard deviation of the MSE for each model across different ground truth functions. As expected, each fixed-activation model achieves zero error only when its activation matches the ground truth. In contrast, `Flex-Act` consistently converges to the optimal function, matching or outperforming all baselines in every setting without any additional computational overhead. This advantage helps circumvent the need to train different models for each activation functions and provides both interpretability and efficiency in training.

**Table 1.** MSE (mean $\pm$ std) between learned and ground-truth activations. **Bold** denotes the best result per column; $\dagger$ denotes the second best. Rows shaded in blue-gray correspond to our proposed `Flex-Act`.

| Model | ReLU | Sigmoid | Tanh | LeakyReLU | Identity |
|---|---|---|---|---|---|
| *Our method* | | | | | |
| `Flex-Act` ($\alpha$=0.3) | $0.0001 \pm 0.0002^\dagger$ | $0.0011 \pm 0.0006^\dagger$ | $0.0001 \pm 0.0002^\dagger$ | $0.0001 \pm 0.0002^\dagger$ | $\mathbf{0.0000 \pm 0.0000}$ |
| `Flex-Act` ($\alpha$=0.0) | $0.0001 \pm 0.0001^\dagger$ | $0.0059 \pm 0.0073$ | $0.0021 \pm 0.0028$ | $0.0001 \pm 0.0002^\dagger$ | $\mathbf{0.0000 \pm 0.0000}$ |
| *Fixed-activation baselines* | | | | | |
| `ReLU` | $\mathbf{0.0000 \pm 0.0000}$ | $0.0059 \pm 0.0072$ | $0.4328 \pm 0.4287$ | $0.0004 \pm 0.0007$ | $4.1675 \pm 6.7199$ |
| `Sigmoid` | $2.1360 \pm 3.9912$ | $\mathbf{0.0000 \pm 0.0000}$ | $0.4020 \pm 0.4536$ | $2.1365 \pm 3.9910$ | $6.3056 \pm 6.5772$ |
| `Tanh` | $2.2941 \pm 3.9616$ | $0.0141 \pm 0.0119$ | $\mathbf{0.0000 \pm 0.0000}$ | $2.2909 \pm 3.9623$ | $4.2737 \pm 4.7666$ |
| `LeakyReLU` | $0.0004 \pm 0.0007$ | $0.0059 \pm 0.0072$ | $0.4286 \pm 0.4227$ | $\mathbf{0.0000 \pm 0.0000}$ | $4.0787 \pm 6.5640$ |
| `Identity` | $0.5209 \pm 0.4659$ | $0.0078 \pm 0.0061$ | $0.0985 \pm 0.0798$ | $0.5105 \pm 0.4566$ | $\mathbf{0.0000 \pm 0.0000}$ |

Importantly, the variant without regularization (`Flex-Act` with $\alpha = 0$) performs significantly worse when the ground truth is `Sigmoid` or `Tanh`. This confirms our earlier hypothesis that unbounded activations (e.g., ReLU) dominate gradient flow during training, leading to biased selection without proper correction.

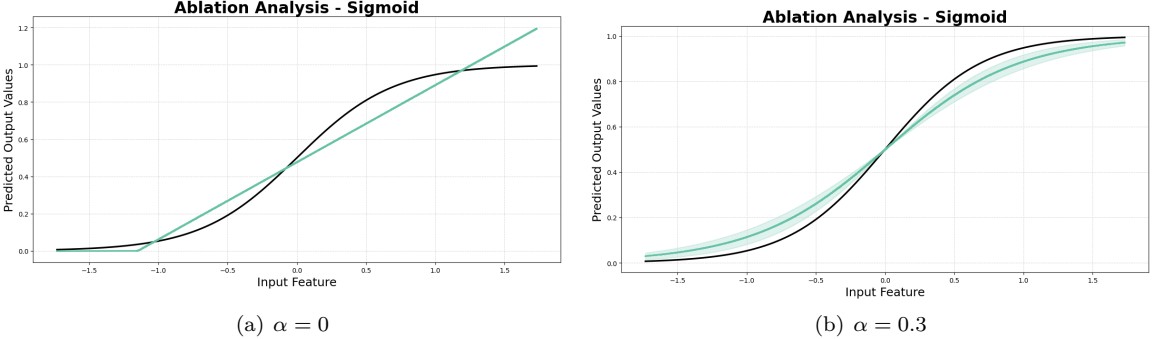

(a) $\alpha = 0$        (b) $\alpha = 0.3$

**Figure 1.** Effect of KL-based regularization on activation selection. Without regularization, the model biases towards unbounded activation functions such as ReLU. With regularization, convergence improves significantly.

**Effect of Regularization.** Figure 1 shows the selection dynamics with and without gradient-based regularization. Without it ($\alpha = 0$), the model lingers on ReLU or LeakyReLU due to their unbounded gradients. With regularization ($\alpha = 0.3$), the model rapidly discovers and locks onto the true activation. A more detailed study is presented in Appendix B.

## 4.2 Scaling to Real-World Benchmarks

To evaluate whether the benefits of `Flex-Act` generalize beyond controlled synthetic setups, we conduct experiments on the CIFAR-10, CIFAR-100 image classification task using standard convolutional architectures, and on text classification settings on the GLUE benchmark using BERT-base models.

In the first setting, we integrate `Flex-Act` into deep residual networks (ResNet18 and ResNet34) and assess whether discrete activation selection can improve accuracy or at least match the performance of carefully

**Table 2.** CIFAR-10 and CIFAR-100 classification accuracy averaged over 10 independent runs (mean $\pm$ std). $p$-values computed via paired $t$-test against the fixed baseline (ReLU). **Bold** = best result; underline = second best. Rows shaded in blue-gray correspond to our proposed `Flex-Act`.

| Model | CIFAR-10 | | CIFAR-100 | |
|---|---|---|---|---|
| | Acc. (%) | $p$-value | Acc. (%) | $p$-value |
| *ResNet-18* | | | | |
| **Baseline (ReLU)** | $95.35 \pm 0.06$ | - | $77.72 \pm 0.11$ | - |
| + PReLU (He et al., 2015) | $94.94 \pm 0.51$ | 0.154 | $77.62 \pm 0.38$ | 0.643 |
| + Swish-$\beta$ (Ramachandran et al., 2018) | $94.76 \pm 0.58$ | 0.075 | $77.38 \pm 0.46$ | 0.237 |
| + aff({id, ReLU, tanh}) (Manessi & Rozza, 2018) | $94.65 \pm 0.05$ | $<0.001$ | $76.79 \pm 0.59$ | 0.019 |
| + AReLU (Hu et al., 2022) | $94.39 \pm 0.47$ | 0.009 | $76.92 \pm 0.11$ | $<0.001$ |
| + `Flex-Act` (penultimate) | $\underline{95.52} \pm 0.05$ | 0.012 | $\underline{79.22} \pm 0.11$ | $<0.001$ |
| + `Flex-Act` (all layers) | $\mathbf{95.68} \pm 0.05$ | $<0.001$ | $\mathbf{79.29} \pm 0.11$ | $<0.001$ |
| *ResNet-34* | | | | |
| **Baseline (ReLU)** | $95.31 \pm 0.06$ | - | $78.62 \pm 0.11$ | - |
| + PReLU (He et al., 2015) | $94.95 \pm 0.57$ | 0.212 | $78.76 \pm 0.11$ | 0.312 |
| + Swish-$\beta$ (Ramachandran et al., 2018) | $94.93 \pm 0.52$ | 0.193 | $78.72 \pm 0.11$ | 0.469 |
| + aff({id, ReLU, tanh}) (Manessi & Rozza, 2018) | $94.72 \pm 0.50$ | 0.074 | $78.26 \pm 0.11$ | 0.022 |
| + AReLU (Hu et al., 2022) | $94.45 \pm 0.45$ | 0.013 | $77.96 \pm 0.11$ | $<0.001$ |
| + `Flex-Act` (penultimate) | $\underline{95.48} \pm 0.05$ | 0.018 | $\underline{79.32} \pm 0.11$ | $<0.001$ |
| + `Flex-Act` (all layers) | $\mathbf{95.75} \pm 0.05$ | $<0.001$ | $\mathbf{79.42} \pm 0.11$ | $<0.001$ |

hand-tuned fixed activations when added ad-hoc in the penultimate layer of the neural network, and in all layers of the network. We train all models for 200 epochs using SGD with momentum (0.9), batch size 128, and a cosine annealing learning rate schedule starting at 0.1. `Flex-Act` is applied to the respective layers of the network with the same candidate set of activations as in our synthetic setup. Our results are depicted in Table 2 where we note the `Flex-Act` in both of its variants (penultimate and all-layers version) improves upon their respective baseline metrics and are statistically significant with a p-value < 0.05. `Flex-Act` maintains or slightly improves performance over strong baselines, while offering an interpretable advantage: it can recover the identity function or fallback to ReLU if no better choice exists. These results suggest that `Flex-Act` is not only robust to function mismatch in toy tasks but also compatible with deeper architectures trained on real data, reinforcing its viability as a drop-in module for general-purpose neural network training. We also compare against other baselines such as Manessi & Rozza (2018) which considers a convex combination of activation functions and Hu et al. (2022) which adds additional parameters to the base activation function increasing the range of the function along with others such as PReLU and Swish-$\beta$. Additional analysis of performance benchmarking and overhead of our models in comparison to the baselines is depicted in Appendix C.

Table 3 reports results on the GLUE benchmark (Wang et al., 2018) using BERT as the backbone, comparing a fixed GeLU baseline against two variants of our proposed `Flex-Act` one in which the learnable activation is applied only to the penultimate layer, and one in which it replaces activations in all transformer layers. Both variants consistently outperform the GeLU baseline across the majority of tasks. `Flex-Act` applied to the penultimate layer alone yields a macro-average improvement of +1.34 points, with particularly strong gains on CoLA (+3.55 Matthews correlation) and MRPC accuracy (+1.71). Extending `Flex-Act` to all layers further improves performance, achieving the best result on almost every reported task and a macro-average of 83.81, representing a gain of +1.67 points over the baseline. Notably, the largest single-task improvement is observed on WNLI (+5.63 accuracy), a task known to be challenging for standard fine-tuning (Levesque et al., 2012), suggesting that adaptive activations may provide a useful inductive bias for tasks requiring lexical inference. Overall, these results demonstrate that replacing fixed activation functions with `Flex-Act` provides a consistent and broadly applicable improvement to BERT fine-tuning across diverse classification tasks, without any significant increase in inference cost.

**Table 3.** GLUE benchmark results for BERT with a fixed GeLU baseline versus `Flex-Act` variants. Each cell shows the primary metric (top) and the absolute change relative to the baseline (bottom, green = improvement, red = degradation). **Bold** = best per task; underline = second best. Avg. is the macro-average over all available task metrics. Shaded rows are our proposed method.

| | CoLA | MRPC | | STS-B | | QNLI | RTE | WNLI | Avg. |
|---|---|---|---|---|---|---|---|---|---|
| **Model** | Mcc | F1 | Acc | Prs | Spr | Acc | Acc | Acc | |
| **Baseline (ReLU)** | 56.53 | 88.85 | 84.07 | 88.64 | 88.48 | 90.66 | 65.70 | 56.34 | 77.41 |
| | - | - | - | - | - | - | - | - | - |
| + `Flex-Act` (penultimate) | **60.08** | 90.07 | 85.78 | 89.20 | 88.82 | 91.14 | **66.43** | 56.34 | 78.48 |
| | +3.55 | +1.22 | +1.71 | +0.56 | +0.34 | +0.48 | +0.73 | +0.0 | +1.07 |
| + `Flex-Act` (all layers) | 59.40 | **90.78** | **85.85** | **89.70** | **88.92** | **91.22** | 66.06 | **61.97** | **79.24** |
| | +2.87 | +1.93 | +1.78 | +1.06 | +0.44 | +0.76 | +0.36 | +5.63 | +1.76 |

Mcc = Matthews corr.; Prs/Spr = Pearson/Spearman; Avg. = macro-average over metrics.

## 5 Discussion

Our experiments validate that `Flex-Act` is capable of discovering the optimal activation function in a purely data-driven manner. Even in simple synthetic tasks, this flexibility enables better fit and interpretability than fixed or parameterized alternatives. Importantly, we show that naive use of Gumbel-Softmax leads to biased selection, and that our gradient-norm-based regularizer effectively corrects this issue. The interpretability of `Flex-Act` is a byproduct of its design: discrete selection exposes layer-wise preferences, enabling post hoc analysis and debugging agnostic of the task at hand. While such properties are often overlooked in deep learning systems, we find them critical for understanding and correcting unexpected optimization behavior, and could open up further interesting research avenues into choices of activation functions in deep learning literature. Furthermore, our current regularizer is heuristic and derived from empirical intuition; future efforts could explore alternative formulations grounded in information theory or activation smoothness metrics alleviating the need to tune the regularization hyperparameter, and thus easier to extend across layers.

**Computational Trade-offs.** The training overhead of `Flex-Act` relative to fixed-activation baselines warrants explicit discussion. As reported in Appendix C, the all-layers configuration incurs a ∼50% training latency increase on ResNet-34 compared to the ReLU baseline, while inference overhead remains modest (up to 3.17%). Crucially, the relevant comparison is not a larger model, but the cost of sweeping activation functions across multiple training runs: exhaustively searching over the candidate set would require 40 and 80 separate training runs for ResNet-18 and ResNet-34 models respectively. A 50% increase in a single training run is substantially cheaper than this alternative, particularly when the optimal non-linearity is unknown a priori, and we work with deeper networks.

We further emphasize two important caveats: Firstly, this cost is *training-time only*, as once temperature $\tau$ anneals to near-zero and routing commits to a single activation per layer near the end of training, the selected function can be extracted at the start of inference, reducing serving cost to that of a fixed-activation network with no routing overhead. Secondly, the overhead reflects the parallel evaluation of all candidate branches during exploration is an unavoidable cost for any method that defers activation selection to training time.

The training overhead of `Flex-Act` is therefore best understood as the cost of a single-run hyperparameter search over activation functions, amortised across the training budget. We recommend `Flex-Act` primarily in settings where: (a) retraining multiple models with different fixed activations is infeasible due to compute constraints, (b) the optimal activation is unknown apriori which is true in most cases, or (c) interpretability of layer-wise functional preferences is desired.

# 6 Conclusion

We introduced `Flex-Act`, a framework for discrete activation selection that enables a layer of a neural network to dynamically select its activation function during training. By leveraging the Gumbel-Softmax trick and a novel gradient-norm-based regularizer, our method effectively learns which non-linearity is most appropriate for the task, without requiring architecture-specific tuning or manual hyperparameter sweeps. In controlled experiments where the ground-truth function is known, `Flex-Act` consistently recovered the optimal activation while maintaining interpretability and modularity.

Our results highlight a compelling possibility: the same network architecture can be reused across a diverse range of functional regimes-each requiring distinct activation dynamics-without retraining or redesign. This opens the door to more robust, task-adaptive neural networks where non-linearity is no longer a fixed hyperparameter but a learnable component of the model architecture.

**Broader Impact Statement**

The proposed framework for discrete activation routing has implications for the design and deployment of more adaptive and efficient neural networks. By enabling a single architecture to automatically adjust its activation functions to suit diverse learning tasks, `Flex-Act` reduces the need for manual tuning and hyperparameter searches in this context. This can lower both the computational cost and the barrier to entry for training high-performing models across domains.

We do not foresee direct negative societal impacts of this work. However, as with any general-purpose machine learning method, downstream misuse is always a possibility. To mitigate risks, we encourage the community to pair technical progress with domain-specific oversight when deploying adaptive models in sensitive applications. Furthermore, our method preserves interpretability, which supports transparency and post hoc auditing in high-stakes settings.

**Acknowledgments**

We would like to thank Arun Sai Suggala and Lita Yang for their valuable feedback and insightful discussions at various stages of this project. Their input significantly improved the experimental design and deepened our understanding of the underlying problem.

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

# A    Activation Selection Dynamics and Regression Analysis

In this section, we briefly discuss additional results surrounding our proposed methodology.

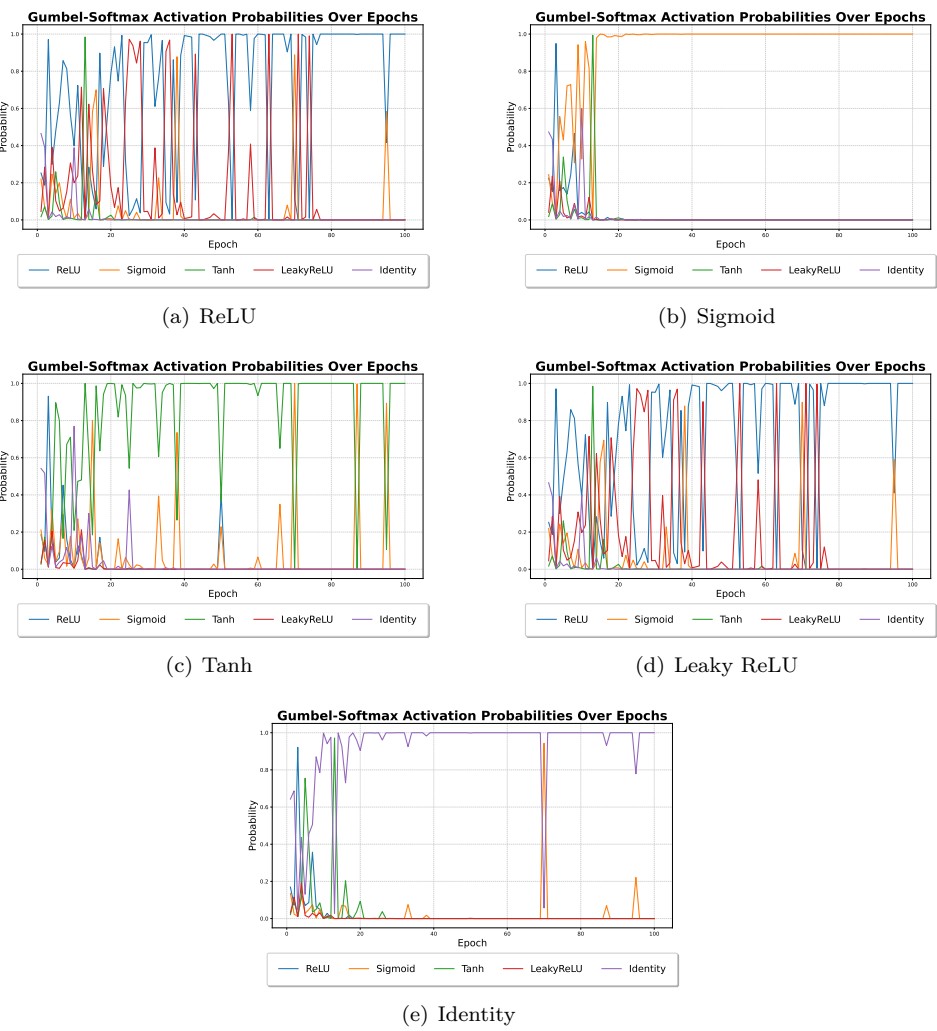

**Figure 2.** Gumbel-Softmax activation selection probabilities over training epochs for various ground-truth activation functions. The model converges to the appropriate non-linearity across tasks with the exception of Leaky ReLU in the above example.

Figure 2 tracks the Gumbel-Softmax probability distribution over time. In each case, the model quickly converges toward the correct activation, sometimes switching transiently between close candidates (e.g., ReLU and LeakyReLU).

To assess the inductive flexibility of learned nonlinearities, we perform an synthetic regression analysis comparing our approach (`Flex-Act`) with standard fixed activation models. The five subfigures in Figure 3 illustrate the predicted output values versus input features for each activation setting.

In each case, the target function is generated using a specific ground truth nonlinearity, while models are trained to regress onto these targets using either a fixed nonlinearity or our adaptive alternative. We visualize predictions from:

- `Flex-Act` with two regularization strengths ($\alpha = 0.0$ and $\alpha = 0.3$),

- standard fixed activation baselines (ReLU, Sigmoid, Tanh, Leaky-ReLU, Identity),

- and the true target output for reference.

(a) ReLU

(b) Sigmoid

(c) Tanh

(d) Leaky-ReLU

(e) Identity

**Figure 3.** Regression analysis comparing our proposed method (`Flex-Act`) against fixed-function models across five ground truth transformations: ReLU, Sigmoid, Tanh, Leaky-ReLU, and Identity. The plots show predicted output values against input features. `Flex-Act` consistently approximates the true functional forms more faithfully, even in the absence of explicit architectural inductive bias.

The results show that `Flex-Act` closely tracks the ground truth across all activation settings:

- For ReLU and Leaky-ReLU, where piecewise-linear structure dominates, our method performs on par with the fixed activations, attaining near-zero approximation error.

- In the case of Sigmoid and Tanh, where saturation and curvature play a larger role, `Flex-Act` exhibits significant robustness, capturing nonlinear trends without overfitting or distortion.

- For the Identity function, fixed nonlinear activations incur substantial bias, whereas `Flex-Act` correctly recovers the linear mapping with zero error.

These visual results complement the quantitative analysis presented in Table 1, wherein `Flex-Act` achieves the lowest or second-lowest mean squared error in every setting. The qualitative agreement further substantiates our claim that learning the activation space dynamically allows for greater generalization and adaptability than committing to a fixed functional form a priori.

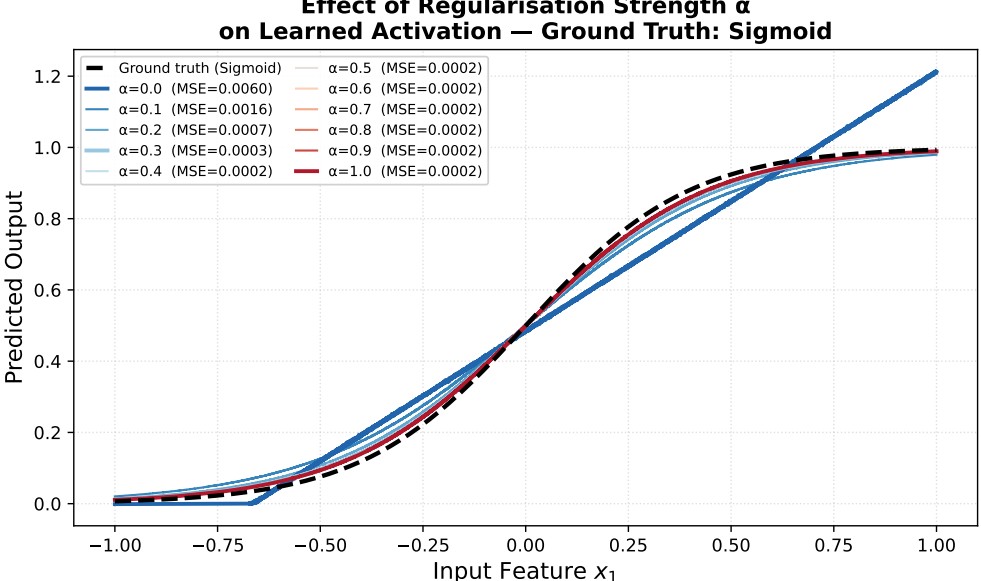

**Figure 4.** Predicted output curves for all $\alpha$ values overlaid against the Sigmoid ground truth (black dashed). The $\alpha = 0$ curve (blue) is piecewise-linear with a visible kink near $x_1 = -0.6$, characteristic of ReLU-dominated routing. All $\alpha \geq 0.1$ curves recover the smooth sigmoid shape; higher $\alpha$ values track the ground truth more tightly, particularly in the saturation region $x_1 < -0.5$. Curves for $\alpha \geq 0.4$ are visually coincident.

## B  Effect of regularization

In this section, we detail the effect of the regularization factor ($\alpha$) on the convergence and performance of the algorithm. Using the same fixed learning as our main experiment, we vary the regularization strengths ($\alpha in [0, 0.1, 0.2, 0.3, 0.4, 0.5, 0.6, 0.7, 0.8, 0.9, 1]$).

**Table 4.** Final MSE and selected activation as a function of regularisation strength $\alpha$ (ground truth: Sigmoid, mean over 3 seeds). For $\alpha \geq 0.1$, the router assigns probability 1.00 to Sigmoid in all runs.

| $\alpha$ | MSE ($\times 10^{-4}$) | Selected Activation |
|---|---|---|
| 0.0 | 60.0 | diffuse (ReLU + LeakyReLU dominant) |
| 0.1 | 16.0 | Sigmoid |
| 0.2 | 7.0 | Sigmoid |
| 0.3 | 3.0 | Sigmoid |
| 0.4 | 2.0 | Sigmoid |
| 0.5 | 2.0 | Sigmoid |
| 0.6 | 2.0 | Sigmoid |
| 0.7 | 2.0 | Sigmoid |
| 0.8 | 2.0 | Sigmoid |
| 0.9 | 2.0 | Sigmoid |
| 1.0 | 2.0 | Sigmoid |

We investigate how the regularisation coefficient $\alpha$ in Eq. 3 governs both the convergence behaviour of the activation-routing mechanism and the quality of the learned function. Using the same synthetic regression setup as Section 4 (ground truth: Sigmoid, 100 epochs, 3 seeds per condition), we sweep $\alpha \in \{0.0, 0.1, 0.2, 0.3, 0.4, 0.5, 0.6, 0.7, 0.8, 0.9, 1.0\}$ and report both the predicted output curves and the final routing distributions. Results are shown in Figures 4, 5, and 6.

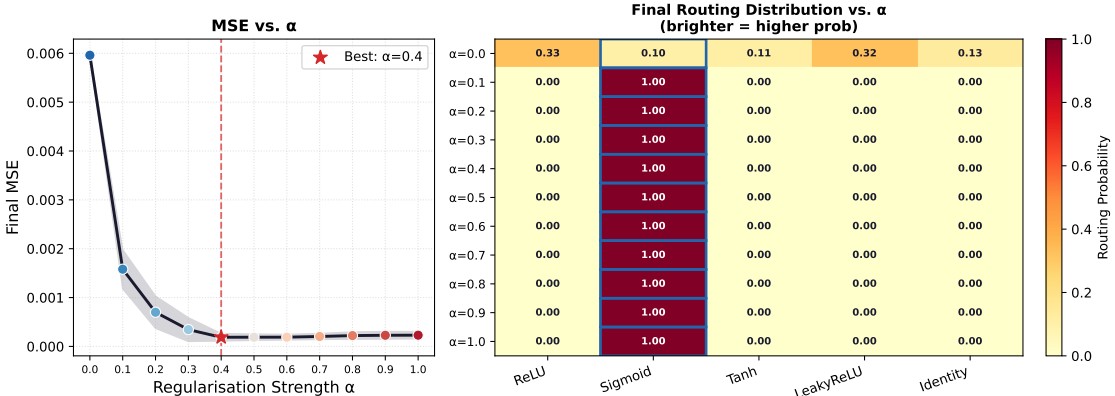

**Figure 5. Left:** Final MSE as a function of $\alpha$, averaged over 3 seeds (shaded band = $\pm 1$ std). The curve shows a sharp elbow between $\alpha = 0$ and $\alpha = 0.1$, confirming that any non-zero regularisation weight decisively corrects the routing bias. MSE reaches its minimum at $\alpha = 0.4$ (red star) and plateaus for all larger values, indicating the regulariser ceases to exert gradient pressure once the routing has converged to the correct activation (Proposition D.6). **Right:** Heatmap of the final routing probability assigned to each candidate activation, for each $\alpha$. At $\alpha = 0$ probability is diffuse across ReLU, LeakyReLU, and Identity - the three activations with $\mathbb{E}[|\sigma'(h)|] \approx 1.0$ - confirming the structural bias of Corollary D.3. For all $\alpha \geq 0.1$, the Sigmoid column (outlined in blue) holds probability 1.00, demonstrating that the correction is both decisive and robust to the choice of $\alpha$.

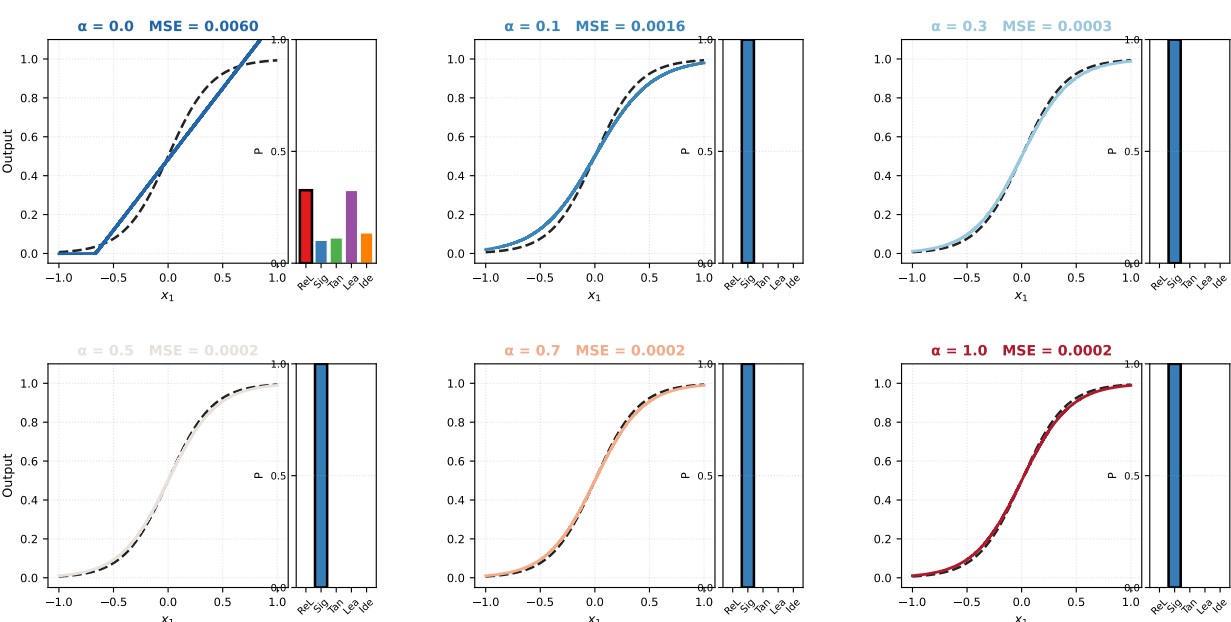

**Figure 6.** Small-multiple view of predicted output (main panel, solid curve vs. ground-truth dashed) and final routing distribution (inset bar chart) for $\alpha \in \{0.0, 0.1, 0.3, 0.5, 0.7, 1.0\}$. At $\alpha = 0.0$ the inset shows a diffuse distribution with ReLU and LeakyReLU dominant; the prediction is piecewise-linear. From $\alpha = 0.1$ onward the Sigmoid bar fills completely (routing probability = 1.00) and the prediction curve tightens progressively onto the ground truth. The visual indistinguishability of the curves for $\alpha \geq 0.5$ is consistent with the MSE plateau in Figure 5 (left).

**Without regularisation the router is biased and does not recover the correct activation.** At $\alpha = 0$, the Gumbel-Softmax router converges to a *diffuse* routing distribution: ReLU (0.33), LeakyReLU (0.32), Sigmoid (0.10), Tanh (0.11), Identity (0.13), as shown in the top row of the routing heatmap (Figure 5,

right). The predicted output is a piecewise-linear, ReLU-like function with a visible kink at $x_1 \approx -0.6$ (Figure 4, blue curve; Figure 6, top-left panel), achieving MSE $= 6.0 \times 10^{-3}$. This behaviour is precisely what Corollary D.3 predicts: without correction, the routing gradient is dominated by the advantage scores of unbounded activations (ReLU, LeakyReLU, Identity all have $\mathbb{E}[|\sigma'(h)|] \approx 1.0$), causing them to collectively absorb routing probability at the expense of Sigmoid ($\mathbb{E}[|\sigma'(h)|] \leq 0.25$). Crucially, the router does not self-correct over training: the task loss adapts the linear layer to work with the biased activation blend rather than discovering Sigmoid.

**Even a small regularisation weight decisively corrects the bias.** At $\alpha = 0.1$, the routing distribution collapses sharply to Sigmoid $= 1.00$ (all other activations $= 0.00$), and MSE drops to $1.6 \times 10^{-3}$ - a $3.75\times$ improvement. For all $\alpha \geq 0.1$ the Sigmoid column in Figure 5 (right) shows probability 1.00, confirming that the KL correction reliably identifies the correct activation across the full range of regularisation strengths tested. This robustness is by design: the pseudo-probability target $\tilde{p}^{(k)} \propto \exp(-\bar{g}^{(k)}/\tau)$ assigns low probability to high-norm activations (ReLU, LeakyReLU, Identity) and high probability to Sigmoid regardless of the exact value of $\alpha$, provided $\alpha > 0$. The temperature coupling ensures the signal is soft enough during early exploration to not over-commit before the linear layer has adapted.

**MSE improves monotonically up to $\alpha \approx 0.4$, then plateaus.** Figure 5 (left) shows a sharp elbow in the MSE curve between $\alpha = 0.0$ and $\alpha = 0.1$, followed by continued improvement through $\alpha = 0.4$ (best MSE $= 2 \times 10^{-4}$), after which performance plateaus for all $\alpha \in [0.4, 1.0]$ at MSE $\approx 2 \times 10^{-4}$. The plateau indicates that once the routing has committed to the correct activation, the primary determinant of MSE is the linear layer's fit to the target - not the strength of the routing correction. The absence of any degradation at large $\alpha$ (e.g., $\alpha = 1.0$ achieves the same MSE as $\alpha = 0.4$) demonstrates that the regulariser does *not* compete destructively with the task loss: once the correct activation is selected, the KL term is satisfied ($p^{(k)} \approx \tilde{p}^{(k)}$, so $\partial \mathcal{L}_{\mathrm{KL}}/\partial \log \pi^{(k)} = \frac{1}{\tau}(p^{(k)} - \tilde{p}^{(k)}) \approx 0$, Proposition D.6) and ceases to exert meaningful gradient pressure on the logits.

**Qualitative improvement in function shape is consistent with quantitative gains.** The small-multiple grid (Figure 6) reveals a clear visual progression. At $\alpha = 0.0$ the predicted curve is piecewise-linear and fails to capture the saturating behaviour of Sigmoid at both tails. At $\alpha = 0.1$ the curve is already smooth and sigmoid-shaped but slightly underestimates the saturation at $x_1 < -0.5$. From $\alpha = 0.3$ onward the predicted curve is visually indistinguishable from the ground truth across the full input range, and the inset routing bar confirms that Sigmoid has monopolised the routing distribution.

**Practical recommendation.** The results suggest that $\alpha \in [0.3, 0.5]$ provides the best combination of routing correction and insensitivity to the exact value chosen: the improvement over $\alpha = 0$ is maximal, MSE is near its floor, and neither too-weak nor too-strong regularisation is a concern. We use $\alpha = 0.3$ throughout the main experiments as a conservative default.

## C   Performance Benchmarking

**Discussion of Computational Characteristics** We evaluated the architectural overhead of the `Flex-Act` framework across three primary dimensions: computational throughput, peak memory utilization, and parameter efficiency.

**1. Latency and Temperature Annealing:** The training latency overhead in `Flex-Act` primarily stems from the computation of a convex combination over the set of candidate activation functions. During the early stages of training, the model explores the functional space by maintaining a soft mixture of activations. As the temperature $\tau$ is annealed toward zero, the Gumbel-Softmax distribution sharpens, causing the model to transition from exploring blends to committing to a specific activation per layer. This process ensures that while training requires evaluating multiple candidate paths, the model eventually converges to a single, distinct choice, minimizing deployment-time overhead.

**2. Memory and Parameter Scalability:** Our results demonstrate that `Flex-Act` introduces negligible parameter overhead, adding only a small set of trainable logits ($\pi$) per layer to manage selection probabilities.

**Table 5.** Computational Overhead Analysis of Flex-Act. Training and inference latencies are measured as milliseconds per step on CIFAR-10. GPU Memory indicates peak allocation during the training phase. "Extra Params" refers to the additional trainable logits introduced by the Gumbel-Softmax selection mechanism.

| Model Architecture | Latency (ms/batch) | | GPU Mem | Extra Params |
|---|---|---|---|---|
| | **Training** | **Inference** | **(MB)** | **(#)** |
| *Baselines (ReLU)* | | | | |
| ResNet18 | 22.599 | 5.61 | 713 | — |
| ResNet34 | 41.529 | 10.284 | 1195 | — |
| *Penultimate* | | | | |
| `Flex-Act`-18-pen | 28.966 | 5.783 | 713 | 5 |
| `Flex-Act`-34-pen | 44.691 | 10.438 | 1196 | 5 |
| *All Layers* | | | | |
| `Flex-Act`-18-all | 40.688 | 5.788 | 1057 | 40 |
| `Flex-Act`-34-all | 66.384 | 10.512 | 1839 | 80 |

For instance, the penultimate configuration adds only 5 parameters regardless of the backbone depth. Memory overhead is more pronounced in the "All Layers" configuration, scaling with the number of layers as the model must store intermediate activations for each candidate function during the training phase to facilitate gradient flow through the routing mechanism.

**3. Interpretability and Functional Discovery:** By exposing activation selection as a discrete, learnable variable, `Flex-Act` provides unique insights into the functional requirements of the network. Unlike parameterized activations that remain within a single functional family, `Flex-Act` can recover entirely different behaviors-such as switching between the sparsity of ReLU and the bounded saturation of Tanh-based purely on data-driven suitability. This allows a single architecture to be reused across diverse functional regimes without manual hyperparameter sweeps or architectural redesign.

## D  Gradient Dynamics and Bias in Discrete Activation Routing

This section provides a complete derivation of the gradients governing the proposed routing mechanism and formally characterizes an inherent scale-induced bias. We further show why standard techniques such as gradient clipping or normalization fail to correct this bias, and derive our correction term in closed form.

### D.1  Setup

At layer $i$, we define the routed activation:

$$x_i = \sum_{j=1}^{K} p_i^{(j)} \sigma^{(j)}(h_i), \tag{5}$$

$$h_i = W_i x_{i-1} + b_i, \tag{6}$$

where the routing probabilities are given by the Gumbel-Softmax:

$$p_i^{(j)} = \frac{\exp\left((\log \pi_i^{(j)} + g_i^{(j)})/\tau\right)}{\sum_{l=1}^{K} \exp\left((\log \pi_i^{(l)} + g_i^{(l)})/\tau\right)}, \qquad g_i^{(j)} \sim \text{Gumbel}(0,1). \tag{7}$$

Let $a_i^{(j)} := \sigma^{(j)}(h_i) \in \mathbb{R}^d$ denote the output of branch $j$, and let $\delta_i := \partial \mathcal{L}/\partial x_i$ denote the upstream gradient, and $z_i^{(k)} := (\log \pi_i^{(k)} + g_i^{(k)})/\tau$.

### D.2 Proposition 1: Routing Gradient

**Proposition 1.** *The gradient of the loss with respect to the routing logits satisfies:*

$$\frac{\partial \mathcal{L}}{\partial \log \pi_i^{(k)}} = \frac{1}{\tau} p_i^{(k)} \left\langle \delta_i, \, a_i^{(k)} - x_i \right\rangle. \tag{8}$$

*Proof.* By the chain rule,

$$\frac{\partial \mathcal{L}}{\partial z_i^{(k)}} = \sum_{j=1}^{K} \frac{\partial \mathcal{L}}{\partial p_i^{(j)}} \frac{\partial p_i^{(j)}}{\partial z_i^{(k)}}. \tag{9}$$

Since $x_i = \sum_j p_i^{(j)} a_i^{(j)}$, we have:

$$\frac{\partial \mathcal{L}}{\partial p_i^{(j)}} = \left\langle \delta_i, a_i^{(j)} \right\rangle. \tag{10}$$

The softmax Jacobian gives:

$$\frac{\partial p_i^{(j)}}{\partial z_i^{(k)}} = p_i^{(j)} (\mathbf{1}_{j=k} - p_i^{(k)}). \tag{11}$$

Combining,

$$\frac{\partial \mathcal{L}}{\partial z_i^{(k)}} = \sum_{j=1}^{K} \left\langle \delta_i, a_i^{(j)} \right\rangle p_i^{(j)} (\mathbf{1}_{j=k} - p_i^{(k)}) \tag{12}$$

$$= p_i^{(k)} \left( \left\langle \delta_i, a_i^{(k)} \right\rangle - \sum_{j=1}^{K} p_i^{(j)} \left\langle \delta_i, a_i^{(j)} \right\rangle \right). \tag{13}$$

Using $x_i = \sum_j p_i^{(j)} a_i^{(j)}$, we obtain:

$$\frac{\partial \mathcal{L}}{\partial z_i^{(k)}} = p_i^{(k)} \left\langle \delta_i, a_i^{(k)} - x_i \right\rangle. \tag{14}$$

Since $\partial z_i^{(k)} / \partial \log \pi_i^{(k)} = 1/\tau$, the result follows. $\qquad \square$

### D.3 Corollary 1: Scale-Induced Selection Bias

Define the advantage score:

$$A_i^{(k)} := \left\langle \delta_i, a_i^{(k)} \right\rangle - \langle \delta_i, x_i \rangle. \tag{15}$$

Then:

$$\frac{\partial \mathcal{L}}{\partial \log \pi_i^{(k)}} \propto p_i^{(k)} A_i^{(k)}. \tag{16}$$

Thus, routing decisions are governed by the inner product $\langle \delta_i, a_i^{(k)} \rangle$, which scales with the magnitude of $a_i^{(k)}$.

For activations with larger output variance (e.g., ReLU), this term is systematically larger in expectation. For instance, if $h \sim \mathcal{N}(0, \sigma_h^2)$ and $\delta_i > 0$, then:

$$\mathbb{E}[\delta_i \cdot \mathrm{ReLU}(h)] \propto \sigma_h, \qquad \mathbb{E}[\delta_i \cdot \sigma(h)] \leq |\delta_i|. \tag{17}$$

This establishes a structural bias favoring high-magnitude activations, independent of optimization hyperparameters.

### D.4 Why Gradient Clipping and Normalization Fail

**Gradient Clipping.** Let $\tilde{\delta}_i = c \cdot \delta_i$ for some scalar $c > 0$. Then:

$$\langle \tilde{\delta}_i, a_i^{(k)} \rangle = c \langle \delta_i, a_i^{(k)} \rangle, \tag{18}$$

which preserves the ordering of advantage scores. Hence, clipping rescales gradients but does not alter branch selection.

**Output Normalization.** Normalizing branch outputs,

$$\hat{a}_i^{(j)} = \frac{a_i^{(j)}}{\|a_i^{(j)}\|}, \tag{19}$$

removes magnitude bias but also removes task-relevant information, impairing function approximation.

### D.5 Regularization via Gradient-Norm-Based Targets

Define branch sensitivity:

$$g_i^{(j)} := \|\nabla_h \sigma^{(j)}(h)\|_2. \tag{20}$$

We construct pseudo-target probabilities:

$$\tilde{p}_i^{(k)} = \frac{\exp(-\bar{g}_i^{(k)}/\varphi)}{\sum_{l=1}^{K} \exp(-\bar{g}_i^{(l)}/\varphi)}, \tag{21}$$

where $\bar{g}_i^{(k)}$ denotes the batch-averaged gradient norm.

### D.6 Proposition 2: KL Correction Gradient

**Proposition 2.** *Let $\mathcal{L}_{\mathrm{KL}} = \mathrm{KL}(\tilde{p}_i \| p_i)$, with $\tilde{p}_i$ treated as a fixed target. Then:*

$$\frac{\partial \mathcal{L}_{\mathrm{KL}}}{\partial \log \pi_i^{(k)}} = \frac{1}{\tau} \left( p_i^{(k)} - \tilde{p}_i^{(k)} \right). \tag{22}$$

*Proof.* Using $\partial \mathcal{L}_{\mathrm{KL}}/\partial p_i^{(k)} = -\tilde{p}_i^{(k)}/p_i^{(k)}$ and the softmax Jacobian, we obtain:

$$\frac{\partial \mathcal{L}_{\mathrm{KL}}}{\partial z_i^{(k)}} = p_i^{(k)} - \tilde{p}_i^{(k)}. \tag{23}$$

The result follows from $\partial z_i^{(k)}/\partial \log \pi_i^{(k)} = 1/\tau$. $\square$

### D.7 Final Gradient

Combining both terms:

$$\frac{\partial \mathcal{L}}{\partial \log \pi_i^{(k)}} = \frac{1}{\tau} p_i^{(k)} \left\langle \delta_i, a_i^{(k)} - x_i \right\rangle + \frac{\alpha}{\tau} \left( p_i^{(k)} - \tilde{p}_i^{(k)} \right). \tag{24}$$

The first term reflects task-driven selection but is biased by activation scale. The second term provides a branch-specific correction that counteracts this bias.

**Table 6.** Comparison of regularisation strategies for matching Sigmoid ground-truth activations. MSE reported as mean $\pm$ std over multiple runs. **Bold** = best result. Our method achieves a 5.4$\times$ reduction over the no-correction baseline.

| Method | MSE (Sigmoid GT) $\downarrow$ |
|---|---|
| No correction *(baseline)* | $0.00590 \pm 0.00000$ |
| Logit grad clipping | $0.00590 \pm 0.00000$ |
| KL + clipped norms | $0.04235 \pm 0.00078$ |
| Output normalisation | $0.35629 \pm 0.00938$ |
| KL + GRAD NORM *(ours)* | $\mathbf{0.00110 \pm 0.00060}$ |

## D.8 Experimental Validation

We evaluate the proposed correction on a synthetic regression task with sigmoid ground truth. This setting is particularly challenging due to the small gradient magnitude of the sigmoid function. The results align precisely with the theoretical analysis:

- Gradient clipping has no effect on routing, confirming that it preserves the ordering of advantage scores.

- Clipping gradient norms before constructing targets destroys relative ordering, leading to degraded performance.

- Output normalization distorts the functional signal, resulting in severe degradation.

- The proposed method achieves consistent and stable convergence by correcting the routing bias without altering the functional representation.

These findings validate that the proposed correction addresses a structural issue in discrete routing that cannot be resolved by standard optimization techniques.

