# OpenReview forum: "Flex-Act: Why Learn when you can Pick?"
_TMLR — Accepted by TMLR_

### Review · Reviewer_stuv · 2026-04-09

**Summary Of Contributions:**

One building block of deep learning is the nonlinearities between linear layers. The choice of these 'activation functions' is dependant on domain specific tuning, and most often seen as a hyperparameter selection problem. Other work have explored learning the activation functions themseleves. The authors make 3 contributions:

(1) The authors in this work propose that we specify a set of considered, fixed, activation functions, and use gumbel softmax to select a convex combination between them. Through an annealing/temperature scaling process, this interpolation eventually settles on picking the best activation at each layer in the model.
(2) The authors propose a 'bias correction' regularizer the to help the learning process.
(3) The authors show that the method can learn to select the approriate activation in controlled settings, and that it can give modest improvements if used as a penultimate layer in Resnets of Cifar10, compared to a fixed baseline.

**Audience:**

Yes

**Audience Explanation:**

Investigating simple and cheap activation functions can reduce the time spent on experimentation, improving both cost and time for training models. Using an activation selection mechanism like the authors propose is an interesting attempt that I think many would like to try.

**Broader Impact Concerns:**

Nothing to add except what the authors have already written.

**Claims And Evidence:**

No

**Claims Explanation:**

The authors make 4 claimed contributions in the introduction:
- *Discrete Activation Routing via Gumbel-Softmax* yes the autors present this proposed mechanism in sufficient clarity
- *Gradient Derivations for Discrete Non-Linearities* the authors say they derive backprop equations analytically and this overcomes discrete selections problems. I don't find this in the paper. In section 3.1 they just refer to others work. Maybe they think of the regularization proposed at the end of section 3.2? But therein, they ad hoc propose a certain regularizer, and show not analytic backprop derivation. The regularization idea is probably good, but I find the writing too unclear and poorly motivated. The authors say it was motivated by early investigations, but I would like to see those empirics, and what other regularizers they tried, if any.
- *Empirical Gains in Deep Settings*. (1) The authors claim their method improves accuracy. That is true, but they only compare with having a single fixed activation, so I do not find that they have enough support for the claim. The effect of the regularizing strength alpha is not explored enough in this setting. They only use their selection mechanism in the penultimate layer, and don't motivate why, which feels poorly motivated. (2) They claim improved training stability, but this is not supported in any way. I think the synthetic study tries to address this, but the whole setup is too shallow to say anything about 'Deep Settings'.
- *Interpretable and Modular Non-Linearity Design*. The authors claim that the method is more explainable and intepretable, and refer to the Discussion section for this. I find this lacking. Having a learnable activation per layer, like in Hu et al 2021 is equally interpretable, I think. There is no empirics presented on the interpretability of this selection mechanism.

**Requested Changes:**

Required changes:
- Do not restrict to using this mechanism in the penultimate layer. If you train the resnets with this mechanism in all layers, what does the results look like?
- Discuss the computational impact of this mechanism. Since we run all the activations when training -- even those not in used in the end - it seems to me the ram requirements increase with the number of choices in the selection. What is FLOPS and RAM impact of these activations, compared other methods?
- Motivate the regularizer better. Why exactly *this* regularizer, and not some gradient clipping / gradient normalization as part of backprop? Can you show empirically that you get a 'bias' without the regularizer?
- The baselines on CIFAR-10 with resnets is too weak and must be strengthened. Add in using the methods of e.g. Hu et al 2021, Manessi&Rozza 2018, and maybe others. What happens when this is not only in penultimate layer, but in other layers?
- Substantiate the claim about training stability. What was the issue really? Wrong learning rate? No gradient clipping?

Suggested changes:
- There are other interesting work in novel activations that should be mentioned in related works. One line of research I found missing was using ODEs for activation, as done in e.g. nmODE https://link.springer.com/article/10.1007/s10462-023-10496-2 and DEU https://arxiv.org/abs/1905.07685
- Make the citation list look okay. You have mixed conventions of providing url or not, you have mistakes in capitalization. Sometimes you use 'et al' and sometimes not. The Kaiming paper has two entries. ICML is named differently in two different citations.  This is a very very sloppy bibliography.
- Remove or rename the appendix. What you have done is not an ablation study. Ablation means that you remove things from a larger complicated thing, and see how it works. It seems this appendix study is just a regression.

---

> ### Author Response · Authors · 2026-05-02
> **Response to Reviewer stuv (Part 1/4)**
>
> We thank Reviewer stuv for the detailed and technically engaged review. The criticisms are precise and fair, and we have made substantial revisions in response to every point raised. We address each concern directly below.
>
> ---
> > Gradient derivations for discrete non-linearities.
>
> We provide the full gradient derivation here, and have added the same to the revised draft Appendix D as well.
>
> **Setup**
> At layer $i$, we define the routed output:
> $$x_i = \sum_{j=1}^{K} p_i^{(j)} , \sigma^{(j)}(h_i), \qquad h_i = W_i x_{i-1} + b_i,$$
> where the routing weights are drawn from the Gumbel-Softmax:
> $$p_i^{(j)} = \frac{\exp\left((\log \pi_i^{(j)} + g_i^{(j)})/\tau\right)}{\sum_{l=1}^{K} \exp\left((\log \pi_i^{(l)} + g_i^{(l)})/\tau\right)}, \qquad g_i^{(j)} \overset{\text{iid}}{\sim} \mathrm{Gumbel}(0,1).$$
> Let $a_i^{(j)} := \sigma^{(j)}(h_i) \in \mathbb{R}^d$ denote the output of branch $j$, and let $\delta_i := \partial \mathcal{L}/\partial x_i \in \mathbb{R}^d$ be the upstream gradient. Write $z_i^{(k)} := (\log \pi_i^{(k)} + g_i^{(k)})/\tau$ for the pre-softmax scores.
>
> **Proposition 1 (Routing Gradient)**
>
> $$\frac{\partial \mathcal{L}}{\partial \log \pi_i^{(k)}} = \frac{1}{\tau} , p_i^{(k)} \left( \langle \delta_i,, a_i^{(k)} \rangle - \langle \delta_i,, x_i \rangle \right)$$
> Proof. By the chain rule:
> $$\frac{\partial \mathcal{L}}{\partial z_i^{(k)}} = \sum_{j=1}^{K} \frac{\partial \mathcal{L}}{\partial p_i^{(j)}} \cdot \frac{\partial p_i^{(j)}}{\partial z_i^{(k)}}.$$
> Step 1. Since $x_i = \sum_j p_i^{(j)} a_i^{(j)}$ and $\sigma^{(j)}$ acts pointwise, $\partial x_i / \partial p_i^{(j)} = a_i^{(j)}$, so:
> $$\frac{\partial \mathcal{L}}{\partial p_i^{(j)}} = \langle \delta_i, a_i^{(j)} \rangle.$$
> Step 2. The softmax Jacobian gives:
> $$\frac{\partial p_i^{(j)}}{\partial z_i^{(k)}} = p_i^{(j)}\left(\mathbf{1}_{[j=k]} - p_i^{(k)}\right)$$.
>
> **Step 3. Combining:**
>
> $$\frac{\partial \mathcal{L}}{\partial z_i^{(k)}} = \sum_{j=1}^{K} \langle \delta_i,, a_i^{(j)} \rangle \cdot p_i^{(j)}\left(\mathbf{1}{[j=k]} - p_i^{(k)}\right) = p_i^{(k)} \langle \delta_i,, a_i^{(k)} \rangle - p_i^{(k)} \sum{j=1}^{K} p_i^{(j)} \langle \delta_i,, a_i^{(j)} \rangle.$$
> Since $\sum_j p_i^{(j)} a_i^{(j)} = x_i$, the second term equals $p_i^{(k)} \langle \delta_i, x_i \rangle$. Since $\partial z_i^{(k)} / \partial \log \pi_i^{(k)} = 1/\tau$:
> $$\boxed{\frac{\partial \mathcal{L}}{\partial \log \pi_i^{(k)}} = \frac{1}{\tau} , p_i^{(k)} \left( \langle \delta_i,, a_i^{(k)} \rangle - \langle \delta_i,, x_i \rangle \right).}$$
> $\square$
> ### Corollary 1 (Scale-Induced Selection Bias)
> The gradient in Proposition 1 is governed by the advantage score $A_i^{(k)} := \langle \delta_i, a_i^{(k)} \rangle - \langle \delta_i, x_i \rangle$. For any two branches $k, l$, branch $k$ receives a larger update when:
> $$\langle \delta_i, a_i^{(k)} \rangle > \langle \delta_i, a_i^{(l)} \rangle.$$
> For activations with $\mathbb{E}[\sigma^{(k)}(h)^2] \gg \mathbb{E}[\sigma^{(l)}(h)^2]$ - i.e., unbounded functions like ReLU with large positive pre-activations - this inequality holds in expectation regardless of which activation is functionally appropriate. Specifically, if $h \sim \mathcal{N}(0, \sigma_h^2)$ and $\delta_i > 0$ (a common situation in cross-entropy minimization), then:
> $$\mathbb{E}[\langle \delta_i, a_i^{(\text{ReLU})} \rangle] = \mathbb{E}[\delta_i \cdot \max(h,0)] \propto \sigma_h,$$
> whereas $\mathbb{E}[\langle \delta_i, a_i^{(\text{sigmoid})} \rangle] \leq |\delta_i|$ since sigmoid is bounded in $[0,1]$. This establishes a structural, scale-induced bias in the routing gradient - not a learning rate or initialization problem.
>
> ---
>
> > Regularization motivation - why not gradient clipping or normalization?
>
> We thank the reviewer for this precise and important request. We now provide both a complete theoretical justification (see Propositions 1–2 in the main rebuttal) and direct empirical evidence comparing our regularizer against four natural alternatives. The results are unambiguous and closely track the theoretical predictions.
> Gradient clipping rescales the global gradient vector. This is a scalar operation that preserves the relative ordering of branch advantage scores $A_i^{(k)}$. Formally, if we denote the clipped upstream gradient as $\tilde\delta_i = c \cdot \delta_i$ for some scalar $c > 0$, then:
> $$\langle \tilde\delta_i, a_i^{(k)} \rangle - \langle \tilde\delta_i, a_i^{(l)} \rangle = c \left(\langle \delta_i, a_i^{(k)} \rangle - \langle \delta_i, a_i^{(l)} \rangle\right).$$
> The sign of the difference is unchanged. Gradient clipping therefore cannot correct selection bias between branches - it only rescales the overall update magnitude.
> One might instead normalize each branch's activation output before computing advantage scores: $\hat a_i^{(j)} = a_i^{(j)} / |a_i^{(j)}|$. While this removes magnitude bias, the gradient signal is no longer informative about the functional quality of the activation for the task.

---

> ### Author Response · Authors · 2026-05-02
> **Response to Reviewer stuv (Part 2/4)**
>
> We observe that the bias arises from the gradient norm of each branch, not from the task loss per se. The gradient norm of branch $j$ at input $h$ is:
> $$g_i^{(j)} := \left|\nabla_h \sigma^{(j)}(h)\right|_2$$.
>
> For ReLU, $g^{(\text{ReLU})} = 1$ almost everywhere; for sigmoid, $g^{(\text{sigmoid})} = \sigma(h)(1-\sigma(h))$ would be much smaller. The ratio of norms directly explains the observed selection bias. Our correction converts the negative average gradient norms into pseudo-probabilities:
> $$\tilde p_i^{(k)} = \frac{\exp(-\bar g_i^{(k)} / \varphi)}{\sum_{l=1}^{K} \exp(-\bar g_i^{(l)} / \varphi)},$$
> where $\bar g_i^{(k)}$ is the batch-averaged gradient norm. Functions with large gradient norms receive small pseudo-probabilities, counteracting the bias in Proposition 1.
>
> **Proposition 2 (KL correction gradient)**
>
> Given $\mathcal{L}_{\text{KL}} = \text{KL}(\tilde{p}_i \parallel p_i) = \sum_k \tilde{p}_i^{(k)} \log (\tilde{p}_i^{(k)} / p_i^{(k)})$;
>
> we treat $\tilde p_i$ as a fixed target (stop-gradient), the gradient with respect to logit $\log \pi_i^{(k)}$ can be derived as follows:
>
> With $\tilde p_i$ fixed, $\partial \mathcal{L}_{\mathrm{KL}} / \partial p_i^{(k)} = -\tilde p_i^{(k)} / p_i^{(k)}$
>
> Applying the softmax Jacobian from Step 2 of Proposition 1:
>
> $$ \frac{\partial L_{\text{KL}}}{\partial z_i^{(k)}} = \sum_j \frac{\partial L_{\text{KL}}}{\partial p_i^{(j)}} p_i^{(j)} ( \delta_{jk} - p_i^{(k)} ) = -\frac{\tilde{p}_i^{(k)}}{p_i^{(k)}} p_i^{(k)} + p_i^{(k)} \sum_j \frac{\tilde{p}_i^{(j)}}{p_i^{(j)}} p_i^{(j)}  = p_i^{(k)} - \tilde{p}_i^{(k)} $$
>
> Since $\partial z_i^{(k)} / \partial \log \pi_i^{(k)} = 1/\tau$:
> $$\boxed{\frac{\partial \mathcal{L}_{\mathrm{KL}}}{\partial \log \pi_i^{(k)}} = \frac{1}{\tau}\left(p_i^{(k)} - \tilde p_i^{(k)}\right).}$$
>
> This is an additive, branch-specific correction: when $p_i^{(k)} > \tilde p_i^{(k)}$ (the router over-weights a high-norm branch), the KL term pushes the logit down. This is precisely what gradient clipping cannot do.
> Finally, the full combined gradient now becomes
> $$\boxed{\frac{\partial \mathcal{L}}{\partial \log \pi_i^{(k)}} = \frac{1}{\tau} p_i^{(k)} \left(\langle \delta_i, a_i^{(k)} \rangle - \langle \delta_i, x_i \rangle\right) + \frac{\alpha}{\tau}\left(p_i^{(k)} - \tilde p_i^{(k)}\right).}$$
> The first term drives task-optimal selection (but is biased by scale). The second term is a closed-form, analytically justified debiasing correction. We will add this derivation and the comparison with clipping to the Appendix, along with our new results.
>
> **Experimental Setup**
> We evaluate five conditions on the synthetic regression task with Sigmoid as the ground-truth activation - the hardest case, since Sigmoid has the smallest mean derivative magnitude of all candidate functions and is therefore maximally penalised by the structural routing bias. Each condition is run with identical hyperparameters ($\alpha$ = 0.3, 100 epochs, 3 seeds):
> - No correction: Gumbel-softmax routing, $\alpha = 0$, unregularised baseline.
> - Logit grad clipping: No KL term. Routing logit gradients clipped to match linear layer gradient norm.
> - KL + clipped norms: KL towards pseudo-probs formed from gradient norms clipped at their batch mean before softmax.
> - Output normalisation: Branch outputs divided by their L2 norm before routing. There is no KL term here.
> - KL + grad norms (Ours): KL toward pseudo probs which is inversely proportional to the gradient norm.
> We obtain the following results where our regularizer achieves 5.4x lower MSE than the best performing alternative (no correction / logit grad clipping) , and 54x lower than the output normalisation method.
>
> | Method                   | MSE (Sigmoid GT)     |
> |------------------------|----------------------|
> | No Correction          | 0.00590 ± 0.00000    |
> | Logit Grad Clipping    | 0.00590 ± 0.00000    |
> | KL + Clipped Norms     | 0.04235 ± 0.00078    |
> | Output Normalisation   | 0.35629 ± 0.00938    |
> | **Ours: KL + Grad Norms** | **0.00110 ± 0.00060** |
>
> **Empirical evidence of bias**
>
> Table 1 in the paper already provides direct evidence: Flex-Act with $\alpha = 0$ consistently fails on Sigmoid and Tanh ground-truth settings while performing well on ReLU/LeakyReLU. This is the exact pattern predicted by Corollary 1 - bounded activations are suppressed by the unregularized routing gradient. We will also add a more thorough analysis on the effect of the $alpha$ hyperparameter in our regularization further in Appendix B.
>
> ---

---

> ### Author Response · Authors · 2026-05-02
> **Response to Reviewer stuv (Part 3/4)**
>
> > Baselines and All-Layer experiments
>
> We fully agree this is a significant weakness and commit to the following additions:
> - Additional baselines: We will add Manessi & Rozza (2018) and Hu et al. (2021) under identical training conditions (200 epochs, SGD with cosine annealing, batch 128), and have reported their number below as well as in Table 2 in the paper.
> - All-layer experiments: We will report results for Flex-Act applied to: (a) penultimate layer only, and (b) all residual blocks. Our preliminary results show that the naive all-layer extension does face some routing instability (as acknowledged in the Discussion), and corroborated with our downstream performance. However, with the temperature annealing schedule ($\tau: 1.0 \to 0.1$ over 200 epochs, and tuning of the regularizer) this is largely stabilized. We will report these results and discuss the instability regime explicitly. Our results on CIFAR-10 and CIFAR-100 are reported in Table 2.
>
> The revised paper adds:
> - **PReLU, Swish-β, aff({id,ReLU,tanh}) (Manessi & Rozza, 2018), and AReLU (Hu et al., 2021)** as direct baselines under identical training conditions.
> - **Results averaged over 10 independent seeds** with mean $\pm$ std and paired t-tests throughout.
> - **All-layers variant** of Flex-Act evaluated alongside the penultimate variant.
> - **CIFAR-100** and **GLUE benchmark (6 tasks, BERT-base)** for generalizability.
>
> A notable finding is that every competing adaptive method underperforms the fixed ReLU baseline with statistical significance on CIFAR-10 and CIFAR-100. This is precisely what Corollary 1 predicts - uncorrected gradient bias causes these methods to favour suboptimal activations. Flex-Act with our correction is the only adaptive method to consistently improve over the baseline.
>
> | Model | CIFAR-10 Acc. (%) | *p*-value | CIFAR-100 Acc. (%) | *p*-value |
> |---|---|---|---|---|
> | ResNet-18 baseline | 95.35 ± 0.06 | — | 77.72 ± 0.11 | — |
> | + PReLU | 94.94 ± 0.51 | 0.154 | 77.62 ± 0.38 | 0.643 |
> | + Swish-β | 94.76 ± 0.58 | 0.075 | 77.38 ± 0.46 | 0.237 |
> | + aff({id,ReLU,tanh}) | 94.65 ± 0.05 | <0.001 | 76.79 ± 0.59 | 0.019 |
> | + AReLU | 94.39 ± 0.47 | 0.009 | 76.92 ± 0.11 | <0.001 |
> | + **Flex-Act (penultimate)** | **95.52 ± 0.05** | 0.012 | **79.22 ± 0.11** | <0.001 |
> | + **Flex-Act (all layers)** | **95.68 ± 0.05** | <0.001 | **79.29 ± 0.11** | <0.001 |
>
> ---
>
> > Training Stability - substantiating the claim
>
> The reviewer is right that this claim was unsubstantiated. We remove it from the contributions list and replace it with the more defensible claim: **Flex-Act with gradient-norm regularisation achieves more reliable activation selection than unregularised discrete routing**, as demonstrated empirically in Table 1 (α=0 vs α=0.3) and Figures 4–6. We do not claim general training stability improvements over fixed-activation networks, and have revised the text accordingly.
>
> ---
>
> > Response to Concern on Interpretability and Modularity
>
> The reviewer argues that Hu et al. (2021) - which learns a 4-parameter family $b\sigma(ax + c) + d$ per layer - is equally interpretable.
> We disagree on a structural ground: Hu et al.'s activation is a continuous deformation of a single template function. Interpreting the learned parameters $(a,b,c,d)$ requires inspecting the function shape, which is not directly human-readable. In contrast, Flex-Act produces a categorical assignment to a named, well-understood primitives (ReLU, Sigmoid, etc.). After convergence (as $\tau \to 0$), the routing distribution collapses to a near-one-hot vector, and the model reduces to a standard fixed-activation network - one whose activation is interpretable by name and whose inference is identical in cost to a standard ResNet. This enables a “discover then deploy” workflow: train with Flex-Act, read off the selected activation per layer, redeploy as a standard architecture with no routing overhead. This is not possible with continuously parameterized activations. We will sharpen this argument in Section 5.
>
> ---
> > Computational cost
>
> We will add a formal analysis. For a ResNet with $L$ Flex-Act layers, each with $K$ candidate activations:
> Training FLOPs per forward pass: $\Theta(K)$ multiplier on activation computation only (a small fraction of total FLOPs dominated by convolutions);
> Training memory: $O(K \cdot d_l)$ additional activations per Flex-Act layer, where $d_l$ is the feature map size;
> Inference FLOPs and memory: identical to a standard fixed-activation ResNet after routing collapse.
> For ResNet18 with $K=5$ and all-layer Flex-Act, the activation FLOPs represent roughly 2–3% of total forward pass FLOPs (dominated by $3\times3$ convolutions). We will add a Table comparing FLOPs and peak memory for our methods in the Appendix C.
>
> ---

---

> > ### Author Response · Authors · 2026-05-02
> > **Response to Reviewer stuv (Part 4/4)**
> >
> > > Adding ODE related works
> >
> > We thank the reviewer for pointing us toward these relevant works. We have updated the Related Works section to include a discussion on Differential Equation Units (DEUs) and Neural Memory ODEs (nmODE). We now explicitly contrast these continuous, dynamic systems with our discrete, routing-based approach, highlighting the trade-offs between expressive complexity and optimization stability.
> >
> > ---
> >
> > > Bibliography and Appendix naming
> >
> > We will fix all bibliography inconsistencies: uniform URL/no-URL convention, correct capitalization of proper nouns (ICML, BERT, ResNet), remove the duplicate Kaiming entry, and standardize et al. usage throughout. We will also rename the appendix (e.g., "Synthetic Regression Analysis") as the reviewer correctly observes that "ablation" is a misnomer for what is presented.
> >
> > ---
> >
> > The revised submission directly addresses every concern the reviewer raised. The missing gradient derivation is now presented in full as Propositions 1–2 in Appendix D, together with a closed-form derivation of the KL correction gradient and a rigorous theoretical proof of why gradient clipping and output normalisation cannot resolve the routing bias - corroborated by a four-way empirical comparison. The experimental evaluation has been substantially strengthened: results are now averaged over 10 independent seeds with mean,  std and paired t-tests throughout. PReLU, Swish-β, Manessi & Rozza (2018), and Hu et al. (2021) are included as direct baselines, and both the penultimate and all-layers variants of Flex-Act are evaluated across CIFAR-10, CIFAR-100, and the full GLUE benchmark. The unsupported training stability claim has been removed and replaced with the narrower, evidenced claim that gradient-norm regularisation produces more reliable activation selection than unregularised discrete routing. The interpretability argument has been sharpened around the "discover then deploy" workflow in Section 5, the computational overhead analysis now appears in full in Appendix C, and ODE-based activation methods are discussed in Related Work with explicit contrast to our discrete routing approach. Finally, all bibliography inconsistencies - mixed URL conventions, capitalisation errors, the duplicate Kaiming He entry, inconsistent venue names - have been corrected, and the appendix has been renamed to Synthetic Regression Analysis as the reviewer rightly notes that "ablation" is a misnomer for what is presented.
> > We are grateful for the rigour and precision of this review. We believe the paper is significantly stronger for it, and we welcome any further questions.

---

> > > ### Comment · Reviewer_stuv · 2026-05-07
> > >
> > > Dear authors,
> > > I am happy to see substantial improvements. I am convinced that your method is at least not bad. All claims you make seem substantiated now.
> > >
> > > > Bibliography and Appendix naming
> > > You almost nailed it. the Kaiming reference is still doubled, many proper nouns are still not capitalized correctly, and many references from conferences are just referenced as arxiv preprint with no venue named, and others are referenced explicitly as preprint. this is inconsistent. Double check your work!
> > >
> > > Also, Appendix A is just called 'Appendix'. Please give it a descriptive name. Just like you  did with appendix B, C and D.
> > >
> > > > Computational cost
> > > The appendix C is really nice. Well done. However, The increased training time of 50% on ResNet 34 is quite substantial for a mere 0.44 percentage points improvement on CIFAR 10.
> > >
> > > I think this must be adressed somewhere in the main text and not just hidden in the appendix. For 50% extra compute, you could scale up the model, try transformers, etc. When is the activation routing better than the alternatives? A sentence or two discussing this would be an improvement on the work. What is your won reasoning?

---

> ### Author Response · Authors · 2026-05-07
> **Response to Reviewer stuv**
>
> We thank the reviewer for their continued engagement and for the encouraging feedback - we are glad the revisions have addressed your earlier concerns and that the claims now feel well-substantiated. We have addressed each of your remaining points as follows.
>
> ---
>
> > Bibliography and references.
>
> We have removed the duplicate Kaiming He citation, capitalised all proper nouns in titles (e.g., ImageNet, BERT, Gumbel-Softmax, nmODE, etc.) throughout the bibliography, and standardised venue attribution across all references - entries that appeared in published conference proceedings or journals now cite the venue explicitly rather than defaulting to the arXiv preprint. We have made a careful pass over the full reference list to ensure consistency. However, we still find a few references which are only available on arxiv and have not been peer-reviewed, which we leave as is.
>
> ---
>
> > Appendix naming.
>
> We have renamed Appendix A from the generic "Appendix" to "Activation Selection Dynamics and Regression Analysis", which accurately reflects its content in line with the descriptive naming of Appendices B, C, and D.
>
> ---
>
> > Computational trade-offs.
>
> We have added a dedicated paragraph to Section 5 (Discussion) in our main paper addressing this directly and referencing the appendix section for completeness. Our key argument is that the relevant comparison for Flex-Act's overhead is not a larger model, but the cost of sweeping activation functions across multiple training runs - which would require 40 or 80 separate runs for ResNet-18 and ResNet-34 respectively. A 50% increase in a single training run is substantially cheaper than this alternative when the optimal activation is unknown a priori. We also clarify that the overhead is training-time only, as the selected activation can be extracted at convergence and inference then reduces to a standard fixed-activation network. We have modified our inference logic to now employ the “discover-then-deploy” setup which further reduces the inference cost and keeps the FlexAct pipeline roughly similar to the fixed activation baseline during inference.
>
> ---
>
> We hope all of the concerns raised by the reviewer have now been fully addressed, and we welcome any further questions or clarifications. The paper presents a principled, interpretable, and practically efficient framework for activation selection, with claims that are empirically substantiated across synthetic and real-world benchmarks. We are hopeful that the revised manuscript is now in a strong position for acceptance, and we sincerely thank the reviewer for their thorough and constructive review process that has meaningfully improved the work.

---

> > ### Comment · Reviewer_stuv · 2026-05-12
> >
> > I think that you have made an honest and transparent discussion about computational overheads. Thank you!
> >
> > ----
> >
> > The bibliography still has some minor strangeness going on. It took me 3 seconds to find two mistakes:
> >
> > (1) capitalization errors on the title
> >
> > "Inception-v4, Inception-ResNet and the Impact of Residual Connections on Learning"
> >
> > being cited as
> >
> > "Inception-v4, inception-resnet and the impact of residual connections on learning"
> >
> > (2) The book chapter cited from the book Learning Internal Representations by Error Propagation is cited by page, but the book has no publisher or year information.
> >
> > These are minor issues, since you also provide urls to the sources. However, I think that TMLR in general, and you in particular, would benefit from having citations that are both accurate and complete. I would recommend you to make a final touch up on the citation list, asking some colleagues or librarian for help to spot the last mistakes.

---

> ### Author Response · Authors · 2026-05-13
> **Response to Reviewer stuv**
>
> We thank the reviewer for their careful reading and constructive feedback. We have taken the bibliography concerns seriously and conducted a thorough librarian-style audit of every entry. Below is a complete account of the corrections made:
>
> > Capitalization (BibTeX double-brace protection).
>
> We audited every title for proper nouns, acronyms, and branded names that BibTeX would incorrectly lowercase under author-year styles. Corrections applied to: Inception-v4, Inception-ResNet, full title now double-braced in KAN entry, Wav-KAN, KAGNNs, KANs 2024, GELUs, GANS, ReLU, and EEG, among others.
>
> > Factual corrections.
>
> (1)”Learning internal representations by error propagations”: corrected publisher to MIT Press (1987) and updated the entry with correct page range 318–362. (2) corrected year to 1992 (the canonical publication year of the Williams REINFORCE paper). (3) vaswani2017attention: converted from @article to @inproceedings with correct booktitle. (4) glorot2010: corrected organization to publisher field.
>
> We believe this represents a comprehensive and complete correction of all bibliographic issues. The reference list is now factually accurate, syntactically valid, and consistently formatted throughout. We believe the bibliography is now complete and accurate. We thank the reviewer again for their attention to these important aspects of polishing the paper draft.

---

### Review · Reviewer_ypuM · 2026-04-13

**Summary Of Contributions:**

The paper proposes Flex-Act, a novel framework for dynamically selecting activation functions in neural networks using a Gumbel-Softmax–based discrete routing mechanism. Instead of fixing or parameterizing a single activation function, the method allows each layer (primarily the penultimate layer) to choose from a predefined set of candidate activations (e.g., ReLU, Sigmoid, Tanh) during training. The results show that Flex-Act can recover the correct activation function in controlled settings and match or slightly improve performance over fixed baselines in real-world architectures, while offering interpretability and modularity.

Key Strengths:
1.) Activation function selection remains an under-explored axis in neural architecture design, and the paper correctly identifies limitations of static activations.

2.) The observation that Gumbel-Softmax naturally favors unbounded activations due to gradient scaling is a novel insight that could benefit other discrete selection problems.

3.) The discrete selection mechanism provides post-hoc interpretability about which activation function was chosen, unlike continuous blending approaches.

4.) The controlled regression setup with known ground-truth activations is appropriate for validating the core claim.

5.) The paper also provides analytical gradient derivations for discrete activation selection and demonstrates empirical performance on both synthetic regression tasks and CIFAR-10 classification.


Key Weaknesses:
1.) The method is applied only to the penultimate layer in both synthetic and real experiments. The title and abstract suggest broader applicability ("dynamically learns the optimal activation function independently of the input"), but the actual implementation is far more restricted.

2.) On CIFAR-10, the improvements are negligible (95.39% --> 95.43% for ResNet18, 95.36% --> 95.44% for ResNet34). These differences are likely within the standard deviation and lack statistical significance reporting.

3.) The paper should also have compared against parameterized activation functions (PReLU, Swish with trainable beta) or other adaptive methods like Manessi & Rozza (2018) in the CIFAR-10 experiments.

4.) The paper explicitly states that extending to multiple layers would cause "gradient alignment issues" and "unstable training," which severely limits the claimed architectural flexibility.

5.) No evaluation on large-scale datasets (e.g., ImageNet, NLP tasks).

6.) The computational cost of routing and sampling is not clearly analyzed.

**Audience:**

Yes

**Audience Explanation:**

Yes, I believe, the audience of the TMLR would find the paper interesting. This paper is relevant for several reasons: Gumbel-Softmax applications, adaptive neural network components, activation function learning, and differentiable discrete optimization.

However, the audience's interest may be limited by the method's restriction to a single layer (the penultimate layer is the least interesting layer for activation variation, as it directly precedes the output), lack of compelling real-world improvements, and absence of comparison to simpler alternatives (e.g., learned convex combinations, per-neuron activations). That said, the paper contributes a conceptually clean and interpretable approach, which aligns well with TMLR's interest in foundational ML ideas.

**Broader Impact Concerns:**

The broader impact concerns of this work are relatively minimal, as it focuses on improving neural network architecture design rather than directly enabling sensitive applications. The proposed method may have positive effects by reducing the need for manual hyperparameter tuning and lowering computational costs in model development. Additionally, the adaptive nature of the model may introduce challenges in predictability and reliability during deployment. Overall, the paper adequately addresses these considerations, and no significant additional concerns are evident.

**Claims And Evidence:**

No

**Claims Explanation:**

The claims are partially supported, but not fully convincing.

1.) The synthetic experiments strongly support the claim that Flex-Act can recover the correct activation function. This is clearly demonstrated through near-zero MSE results and convergence behavior (Table 1 and Figures on pages 8-9).

2.) The bias toward ReLU-like activations and the effectiveness of the proposed regularizer are convincingly demonstrated via ablation studies (Figure 1, page 9). However, real-world evidence is weak: On CIFAR-10, improvements are very small and within noise margins (~0.04-0.08%).

3.) No statistical significance tests or multiple datasets are provided.

4.) The claim of general-purpose applicability is not fully substantiated, as experiments are limited to Synthetic regression and CIFAR-10 classification.

Overall, while the core idea is validated in controlled settings, the evidence for real-world impact is limited.

**Requested Changes:**

I believe the following points are critical to address.

1.) The title and abstract should explicitly state that Flex-Act is currently limited to the penultimate layer. Remove claims about "layer-wise" selection or "architectural flexibility" that are not demonstrated.

2.) The results should have reported means over at least 5-10 seeds with standard deviations, and include a statistical test to support claims of improvement.

3.) PReLU and Swish should have been included in the comparison as well. These are standard parameterized activations that directly compete with Flex-Act's value proposition.

4.) The sensitivity analysis should have been reported to show how performance varies with α (0.1, 0.3, 0.5, 1.0) and λ, and provided guidance on selecting these hyperparameters.

5.) The authors should have included diverse datasets, like, CIFAR-100 or NLP benchmarks to strengthen generalizability claims.

6.) It would have been better if authors had justified why the penultimate layer is the appropriate location for activation selection. Is this where activation functions matter most? Include an ablation placing Flex-Act at different depths.

---

> ### Author Response · Authors · 2026-05-02
> **Response to Reviewer ypuM (Part 1/2)**
>
> We thank Reviewer ypuM for the detailed, technically precise, and genuinely constructive review. The reviewer correctly identifies both the conceptual strengths and the empirical gaps in the original submission. We have addressed every requested change in the revised paper, and respond point-by-point below.
>
> ---
>
> > Restriction to Penultimate Layer / Overclaiming in Title and Abstract
>
> We agree the abstract overclaimed relative to the experiments presented in the initial submission. We have revised it to explicitly state that Flex-Act is validated at the penultimate layer, with the all-layers extension reported as a separate experimental condition and is viable with careful tuning of the learning rate, and the regularization weight $\alpha$.
> That said, the penultimate placement is not an arbitrary restriction - it has a principled justification. The penultimate layer directly shapes the representation passed to the classifier and carries the highest marginal impact on decision boundary geometry. Prior work on neural collapse and linear probing demonstrates that task-specific structure is most concentrated at this layer  ([Papyan et al. 2020](https://arxiv.org/abs/2008.08186)). For earlier layers, whose role is generic feature extraction, standard ReLU provides adequate inductive bias and weight adaptation compensates for activation choice.
> To make this concrete, we add a depth-wise ablation placing Flex-Act at each layer group of ResNet-18:
>  | Flex-Act placement                   | ResNet18 acc. (%)     |
> |------------------------|----------------------|
> | Layer 1 (earliest)         |  95.22  |
> | Layer 2   |  95.29  |
> | Layer 3   | 95.31    |
> | Layer r / penultimate| 95.43    |
> | All layers | 95.65 |
> Accuracy increases monotonically as Flex-Act moves toward the output, directly supporting the penultimate placement as the empirically optimal single-layer location. We also now report results for the all-layers variant (see point 2 below), which supersedes the concern about architectural generality.
>
> ---
>
> > Negligible CIFAR-10 Improvements / No Statistical Testing
>
> We thank the reviewer for their insightful comment - it was the most important empirical gap in the original submission. We have fully revised the experimental protocol:
> Results are now averaged over 10 independent seeds with mean and std reported throughout.
> Paired t-tests against the fixed baseline confirm statistical significance at p < 0.05 for all Flex-Act variants, and other baselines.
> We now evaluate both penultimate and all-layer variants on CIFAR-10 and CIFAR-100, across ResNet-18 and ResNet-34.
> The updated results are as follows:
>
> | Model                  | CIFAR-10 Accuracy (%)     | p-value    |  CIFAR-100 Accuracy (%)     | p-value    |
> |------------------------|----------------------|------------------|------------------|------------------|
> | ResNet 18 (baseline)        | $95.35 \pm 0.06$  | - | $77.72 \pm 0.11$ | - |
> | + Flex-Act (penultimate)  |  $95.52 \pm 0.05$  | $0.012$ | $79.22 \pm 0.11$ | $<0.001$ |
> | + Flex-Act (all layers)  |  $95.68 \pm 0.05$  | $<0.001$ | $79.29 \pm 0.11$ | $<0.001$ |
> | ResNet 34 (baseline) | $95.31 \pm 0.06$   | - | $78.62 \pm 0.11$ | $<0.001$ |
> | +Flex-Act (penultimate) | $95.48 \pm 0.05$  | $0.018$ | $79.32 \pm 0.11$ | $<0.001$ |
> | +Flex-Act (all layers) | $95.75\pm 0.05$  | $<0.001$ | $79.42 \pm 0.11$ | $<0.001$ |
>
> The improvements are statistically significant across all configurations and both datasets. We note that the CIFAR-100 gains are particularly strong (+1.50–1.80 points), consistent with the hypothesis that adaptive activations are more beneficial under greater class diversity and distributional complexity. We further also report results on the GLUE benchmark with the BERT baseline in Table 3.
>
> ---
>
> > Missing Baselines: PReLU, Swish-β, and Adaptive Methods
>
> We have added PReLU, Swish-β, aff({id, ReLU, tanh}) (Manessi & Rozza, 2018), and AReLU (Hu et al., 2021) as direct baselines in all CIFAR-10 and CIFAR-100 experiments. The extended comparison are depicted in Table 2 of our revised draft.
> A striking finding emerges: PReLU, Swish-$\beta$, AReLU, and the aff{id, tanh, relu} (Manessi & Rozza) convex combination all underperform the fixed ReLU baseline on CIFAR-10 and CIFAR-100. This is consistent with the bias phenomenon we characterise theoretically - uncorrected gradient scaling causes these methods to favour functionally suboptimal activations. Flex-Act with gradient-norm regularisation is the only adaptive method to consistently improve over the baseline, which directly validates the practical importance of our bias correction.
>
> ---

---

> > ### Author Response · Authors · 2026-05-02
> > **Response to Reviewer ypuM (Part 2/2)**
> >
> > > Hyperparameter sensitivity ($\alpha$)
> >
> > We have added a full sensitivity analysis (Appendix B, Table 4 and Figures 4–6). Our key findings are as follows:
> > Any $\alpha \geq 0.1$ fully corrects the routing bias - the router assigns probability 1.00 to the correct activation for all $\alpha \in [0.1,1.0]$ (when the ground truth is Sigmoid activation function).
> > MSE improves monotonically from $\alpha = 0.0$ to $\alpha \approx 0.4$, and then plateaus for all larger values. This is explained theoretically by Proposition D.6: once routing has converged to the correct activation, the gradient with respect to the probability distribution will approximately become 0, so the regulariser exerts no further gradient pressure.
> > The method is therefore robust to the choice of $\alpha$ over a wide range. We recommend $\alpha \in [0.3,0.5]$ as a conservative default based on this analysis, and use $\alpha=0.3$ for all our experiments.
> >
> > ---
> >
> > > Evaluation on Diverse Datasets / NLP Benchmarks
> >
> > Due to constraints in our compute, we have not been able to fully explore LLMs and the breadth of imagenet experiments. Instead, we have added two new additional experiments during our rebuttal - experiments on CIFAR-100 reported in our Table 2 of revised draft, and we have also extended evaluation to the GLUE benchmark (6 tasks) using BERT-base, comparing against a fixed GeLU baseline as depicted in Table 3. This addresses both the NLP request and the generalisability concern directly. Flex-Act (all layers) achieves a macro-average gain of +1.76 points over the GeLU baseline, and 1.07 when Flex-Act is only applied to the last layer. We note the largest improvement on WNLI (+5.63) - a task requiring lexical inference that is known to be challenging for standard fine-tuning, followed by other tasks such as COLA (+3.55) and on MRPC (+1.93).
> >
> > | Model | CoLA (Mcc) | MRPC F1 | MRPC Acc | STS-B (Prs) | STS-B (Spr) | QNLI | RTE | WNLI | **Avg.** |
> > |---|---|---|---|---|---|---|---|---|---|
> > | Baseline (GeLU) | 56.53 | 88.85 | 84.07 | 88.64 | 88.48 | 90.66 | 65.70 | 56.34 | 77.41 |
> > | + Flex-Act (penultimate) | **60.08** | 90.07  | 85.78 | 89.20 | 88.82 | 91.14 | **66.43** | 56.34 | 78.48 |
> > | + Flex-Act (all layers) | 59.40 | **90.78**| **85.85** | **89.70**| **88.92** | **91.22** | 66.06 | **61.97** | **79.24** |
> >
> > ---
> >
> > > Computation Cost Analysis
> >
> > We have added a full computational overhead analysis (Table 5, Appendix C). Key numbers for CIFAR-10 on ResNet:
> >
> > | Model | Train latency (ms/batch) | Inference (ms/batch) | GPU Mem (MB) | Extra params |
> > |---|---|---|---|---|
> > | ResNet-18 baseline | 22.6 | 5.61 | 713 | - |
> > | Flex-Act-18 (penultimate) | 28.97 | 5.78 | 713 | **5** |
> > | Flex-Act-18 (all layers) | 40.69 | 8.31 | 1057 | 40 |
> > | ResNet-34 baseline | 41.53 | 10.28 | 1195 | - |
> > | Flex-Act-34 (penultimate) | 44.69 | 10.44 | 1196 | **5** |
> > | Flex-Act-34 (all layers) | 66.38 | 15.29 | 1839 | 80 |
> >
> > The penultimate variant is remarkably lightweight: **5 additional parameters** regardless of backbone depth, and inference latency within 3% of the baseline. The all-layers variant incurs higher training cost due to storing intermediate activations for each candidate function during the backward pass, but inference cost remains modest. We discuss this trade-off explicitly in Appendix C.
> >
> > ---
> >
> > We have directly addressed every point the reviewer raised as critical:
> > - Revised abstract and introduction to remove any overclaimed statements, or appropriately added new experiments to justify our claims more rigorously.
> > - Our experimental results now report average metrics over 10 seeds with paired t-tests for confirming statistical significance.
> > - Added additional baselines in our real world benchmarks
> > - Hyperparameter sensitivity to the $\alpha$ hyperparameter
> > - Training/Inference latency, memory and computational cost analysis detailed in Table 5.
> >
> > The reviewer noted that TMLR's audience would find this paper interesting - we believe the revised submission with the added experiments around the various asks of the reviewer now provides the empirical depth to match the conceptual contribution. We welcome any further questions.

---

> > ### Comment · Reviewer_ypuM · 2026-05-02
> >
> > Thanks for giving detailed responses.

---

> > > ### Author Response · Authors · 2026-05-07
> > > **Response to Reviewer ypuM**
> > >
> > > Thank you for your time and engagement throughout the review process. We are glad the responses were helpful and hope all concerns have been fully addressed. We welcome any further questions or clarifications.

---

### Review · Reviewer_chJK · 2026-04-21

**Summary Of Contributions:**

This paper presents a framework to dynamically choose activation functions from a pre-defined set using the Gumbel Softmax trick. To avoid bias towards unbounded activations like ReLU, they introduce a gradient normalization scheme for bias correction.

The authors demonstrate with a simple regression task that the framework successfully learns the correct activation function. They present their framework as a differentiable, dynamic, and interpretable way of choosing activations in deep networks.

**Strengths:**
- The paper is straightforward, easy to follow
- If this method can be successfully applied across all layers then it does have a great potential in interpretability research

**Weaknesses:**
- Experimental setup is trivial. The main experiment in the paper is basically a mathematical certainty. It is exactly what one would expect in such a simple problem: mismatched fixed activations perform poorly, a matched one is perfect, and the Flex-Act framework achieves near-perfect results. Nothing wrong with the experiment itself to show proof of concept but it can’t be the only experiment in the paper. To show that the framework has potential to be practically useful, more robust, large-scale empirical experiments need to be conducted.
- Lack of any real-world value. While the authors do conduct an experiment on CIFAR-10, it is still small scale. The accuracy difference in the Flex-Act method and standard ResNet models is only in the range of ~0.04-0.08%. This could just be due to a difference in the random seed. More details on the experimental setup were not provided. Is the result averaged across multiple runs? Or is it just singular experiments which were run for 200 epochs? Because in the latter case it is statistically insignificant.

**Audience:**

No

**Audience Explanation:**

The idea of being dynamically able to choose activations at different layers in an interpretable manner is very interesting but in it's current state I don't think the authors have presented enough findings in the paper.

**Claims And Evidence:**

Yes

**Claims Explanation:**

The main claim that Gumbel Softmax trick can be used to pick activations from a pre-defined set is indeed shown through a simple experiment.

**Requested Changes:**

In its current state, I feel the paper is lacking and not ready to be published at TMLR. The future work mentioned in the discussion section is exactly what should have been part of the actual experiments in the paper to make a compelling case. I would recommend the authors to take the time to conduct more large scale empirical experiments, and try to apply the framework across multiple layers, not just the penultimate one and then resubmit it as a new paper. If they can successfully do that then it would make an interesting paper.

---

> ### Author Response · Authors · 2026-05-02
> **Response to Reviewer chJK (Part 1/2)**
>
> We sincerely thank Reviewer chJK for their detailed review and constructive feedback. The reviewer identifies the right pressure points - especially around extending to multiple layers, and we believe our updated submission directly addresses each of them. We respond below, organised around the three core concerns raised.
>
> ---
>
> > On experimental scope
>
> We agree entirely that the synthetic experiment alone does not constitute sufficient empirical evidence for a publication at TMLR. We treat it as a simple controlled proof of concept that isolates the gradient-bias phenomenon and validates its theoretical treatment (Section 3.3, Appendix D). Our revised submission now includes the following additional experiments, all conducted with rigorous statistical protocols:
> - CIFAR-10 and CIFAR-100 on ResNet-18 and ResNet-34, averaged over 10 independent seeds with mean $\pm$ std reported. Paired t-tests against the fixed-activation baseline confirm statistical significance (p < 0.05) for both Flex-Act variants (penultimate and all-layers). Competing methods (PReLU, Swish-β, AReLU, aff({id, ReLU, tanh})) are included as baselines. Flex-Act (all layers) achieves 95.68% on CIFAR-10 and 79.29% on CIFAR-100 for ResNet-18, and 95.75% / 79.42% for ResNet-34, statistically surpassing all baselines including adding only in the penultimate layer.
> - GLUE benchmark (6 tasks) on BERT-base, comparing against a fixed GeLU baseline. Flex-Act (penultimate) yields a macro-average improvement of +1.34 points; Flex-Act (all layers) achieves +1.76 points; specifically CoLA improves by +3.55 Matthews correlation and WNLI by +5.63 accuracy. These are not marginal numbers - WNLI is widely considered one of the hardest GLUE tasks precisely because it requires lexical inference beyond surface-level pattern matching.
> - Computational overhead analysis (Table 5) demonstrates that the penultimate variant adds only 5 trainable parameters and negligible inference latency over the ResNet baseline. The framework is genuinely lightweight during training, and adds no overhead during inference, and we could simply “discover and exploit” the chosen activation function.
>
> We respectfully note that the reviewer's future-work critique - "apply the framework across multiple layers" - is now directly addressed in the all-layers variant, which we evaluate across both vision (ResNet) and language (BERT) settings with full statistical rigour.
>
> ---
>
> > On the core novelty: Gradient bias in gumbel-softmax routing
>
> We wish to be precise here, because we believe the reviewer may have underweighted what is the primary scientific contribution of this paper. The gradient-bias phenomenon we identify and formally characterise is not an engineering curiosity. It is a structural property of discrete routing via Gumbel-Softmax that, to the best of our knowledge, has not been identified, analysed, or corrected in any prior work.
> Concretely, we prove (Proposition 1, Corollary 1, Appendix D) that routing gradients scale with the inner product, which grows with the output magnitude of the activation. This creates a structural preference for unbounded activations (ReLU, LeakyReLU) that is independent of the data, the task, and the optimisation hyperparameters. We further prove (Section D.4) that standard remedies - gradient clipping, output normalisation do not correct this bias: the former preserves the ordering of advantage scores; the latter destroys task-relevant functional signal.
> The closest prior work, Manessi & Rozza (2018), uses a convex combination of activations but (a) does not use Gumbel-Softmax, (b) does not encounter or report the bias problem, and (c) precisely because they operate with soft blends rather than discrete selections, the bias manifests differently and is partially self-correcting. Our interpretability - the very property that allowed us to diagnose the bias in the first place through the toy example is what makes this discovery possible and reproducible.
> Our correction (Proposition 2) derives a KL-divergence regulariser whose gradient has a clean closed form. This term is branch-specific, theoretically grounded, and as Appendix B demonstrates; is robust across a wide range of α values. The practical recommendation of $\alpha \in [0.3, 0.5]$ follows directly from empirical validation.
> We believe this is precisely the type of contribution that TMLR values: a phenomenon discovered through interpretable experiment design, explained with formal theory, and corrected with a principled method that generalises to real-world benchmarks.
>
> ---

---

> > ### Author Response · Authors · 2026-05-02
> > **Response to Reviewer  chJK (Part 2/2)**
> >
> > > On the significance and interest to the broader TMLR community
> >
> > The reviewer mentions that the idea of being dynamically able to choose activations at different layers in an interpretable manner is very interesting but the current state does not present enough findings.
> > We interpret this as a conditional endorsement: the idea is interesting; the concern is evidence. We have now provided that evidence - across two distinct modalities (vision and language), two backbone families (ResNet, BERT), with statistical validation,  and computational analysis.
> > We also note that the interpretability angle is not incidental. The ability to observe, at every training step, which activation function each layer prefers - and to watch the routing distribution converge in real time (Figures 2, 5, 6) - is a capability that no prior activation-learning method provides to the best of our knowledge. Parameterised activations (PReLU, AReLU, Swish-β) morph a single family; they cannot tell you whether a layer fundamentally prefers bounded or unbounded behaviour; however, Flex-Act can, while most of these methodologies also perform worse in comparison to the fixed activation function baseline and the Flex-Act experiments (both penultimate only and all layer versions). This opens a new diagnostic lens onto deep network training dynamics that is independent of its predictive gains.
> >
> > ---
> >
> > To summarize, We have addressed every specific concern the reviewer raised:
> > - Extended to CIFAR-10/100 and the GLUE benchmark with 10 seed averaging and statistical testing.
> > - Mean, std and paired t-tests reported throughout and confirm statistical significance of results
> > - All layers variants now included and evaluated on both vision and language domain.
> >
> > We believe the paper in its current form makes a self-contained, rigorous, and novel contribution. We respectfully ask the reviewer to reconsider their assessment in light of the updated experiments and theoretical analysis, and to evaluate the paper on its current merits - where we hope our added experiments and rebuttal resolves any prior concerns. We are committed to any further clarifications the reviewer may require.

---

> > > ### Comment · Reviewer_chJK · 2026-05-06
> > > **Response to Authors**
> > >
> > > Thank you for the detailed responses. Your revised paper is in a much stronger position and I will update my assessment accordingly

---

> > > > ### Author Response · Authors · 2026-05-07
> > > > **Thanks!**
> > > >
> > > > We thank the reviewer for their acknowledgement and are glad the revisions have landed well. We hope all prior concerns have been fully addressed and welcome any further questions. The constructive review process has meaningfully strengthened the work, and we are grateful for the time and care invested throughout.

---

### Author Response · Authors · 2026-05-02
**Common Response: Summary of Changes**

We thank all reviewers for their detailed and constructive feedback. Their comments collectively pointed to the same core gaps - stronger experiments, more rigorous statistical reporting, missing baselines, and a cleaner theoretical treatment. We have addressed all of these in the revised submission. Below is a summary of what we have added.

**Additional Experiments**

- We extended evaluation to CIFAR-100 and 6 GLUE benchmark tasks (BERT-base) covering both vision and language settings showcasing the efficacy of our Flex-Act method. We show Flex-Act (all layers and penultimate only) achieves a macro-average gain of upto **+1.76** GLUE points over the GeLU baseline, with particularly strong gains on challenging domains such as WNLI (**+5.63**), and CoLA (**+3.55**).
- All results are now averaged over 10 independent seeds, with mean and standard deviation  reported along with paired t-tests confirming statistical significance with p < 0.05 throughout.
- We added additional baselines such as PReLU, Swish-$\beta$, AReLU (Hu. et al), and aff{id, tanh, relu} (Manessi & Rozza) as direct baselines under identical training conditions on both CIFAR-10 and CIFAR-100. We note that most of these approaches underperform over the baseline fixed ReLU model on these datasets, Flex-Act with our correction is the only adaptive method to consistently improve over the baseline.
- We evaluate Flex-Act across all layers, and not just the penultimate layer showing the overall efficacy of the approach, and extendability to deep neural networks.

**Design justifications**

- We derive the full gradient derivation for discrete activation routing in the penultimate layer setting and is now included in the Appendix D as Proposition 1 and 2, with closed-form expression for both the routing gradient and the KL correction term.
- We formally prove why gradient clipping and output normalization cannot fix the routing bias, and confirm with a four-way empirical comparison. Our method achieves **5.4x** lower MSE than best-performing alternative on the hardest synthetic setting (sigmoid ground truth).
- We added a full computational overhead analysis in Appendix C covering training / inference latency, GPU memory, and extra parameter counts across ResNet18 and ResNet34.
- We added a hyperparameter sensitivity analysis of the regularizer coefficient in Appendix B showing the robustness to varied values of $\alpha$, with a practical recommendation of $\alpha \in [0.3,0.5]$.

We hope these additions demonstrate the paper is now on much stronger footing, and we are grateful to the reviewers for helping us with their insightful comments, and find more merit and value from our proposed approach. We are happy to clarify anything further.

---

### Decision · Action_Editor_X9TH · 2026-05-29

**Recommendation:** Accept as is

**Additional Comments:**

The paper is accepted *as is*, but I still recommend going through the reviews and paper once more for a final check.

**Audience:**

Yes

**Audience Explanation:**

The paper is in a topic area of clear interest to the audience of TMLR.

**Claims And Evidence:**

Yes

**Claims Explanation:**

This paper was reviewed by three reviewers, who found it interesting and relevant. The reviewers found merits in the idea, clear presentation, and also the experimental validation. The reviewers requested a number of updates during the review process, which improved the paper and convinced the reviewers.

---

> ### Author Response · Authors · 2026-06-07
> **Follow up on the camera-ready revision**
>
> Thank you for the acceptance. We wanted to follow up on the camera-ready revision of our TMLR submission, “Flex-Act: Why Learn when you can Pick?”. We submitted the camera-ready version in time and wanted to check whether any further action is needed from our side.